# Assessing Nowcast Models in the Central Mexico Region Using Radar and GOES-16 Satellite Data

**Diana Islas-Flores [1],\* and Adolfo Magaldi [2]**

1 Programa de Posgrado en Ciencias de la Tierra, Universidad Nacional Autonoma de Mexico, Mexico City 04510, Mexico

2 Escuela Nacional de Estudios Superiores Juriquilla (ENES Juriquilla), Universidad Nacional Autonoma de Mexico, Queretaro 76230, Mexico; adolfo.magaldi@unam.mx

\* Correspondence: daislas@ciencias.unam.mx

**Abstract:** In this study, the nowcast models provided by the Python pySTEPS library were evaluated using radar derived rain rate data and the satellite product Split-Window Difference (SWD) based on GOES-16 data, focusing on central Mexico. Initially, we obtained a characterization of the rainfall that occurred in the region using the radar rain rate and the SWD. Subsequently the nowcasts were evaluated using both variables. Two nowcast models were employed from pySTEPS: Extrapolation and S-PROG. The results indicate that average SWD is below 2.5 K, 90 min before the onset of rainfall events, and, on average, the SWD is 2 K during rainfall events. The results from both nowcast models were accurate and produced similar results. The nowcasts performed better when SWD data were used as input, having an average Probability of Detection (PoD) above 70% and a False Alarm Rate (FAR) reaching 40% for the 15-min prediction. The nowcasts were less accurate using the radar rain rate as input for the 15-min forecast, where the PoD was maximum 70% and FAR reaching 40%. However, these nowcasts were more reliable during well-organized precipitation events. In this work, it was determined that the nowcast models provided by pySTEPS can provide valuable rain forecasts using GOES-16 satellite and radar data for the central Mexico region.

**Keywords:** nowcast; Mexico; weather radar; GOES-16; pySTEPS; Split-Window Difference





## 1. Introduction

Nowcasting models are valuable tools for predicting short-term rainfall events, providing early alerts for potential severe weather in a region. Although these types of models have been implemented around the world, central Mexico has yet to adopt these models, even though the region experiences convective storms during its rainy season, spanning from May to October, which often result in severe flooding. This region it is covered by three radars: one located north of the city of Queretaro (north of Mexico City), one is located in the mountain Cerro Cathedral (west of Mexico City), and the third is located in the center of Mexico City. Additionally, the central Mexico region falls under the coverage of the GOES-R satellite series. These two tools, the radar and satellite data, present an ideal opportunity for the possible implementation of a nowcast system in the region. Such implementation could significantly enhance early detection and preparedness for potential strong convective storms.

Nowcasting models that mainly utilize radar data have been extensively studied and implemented (e.g., [1–8]). An example of this is the Python pySTEPS library, which offers several nowcast models designed specifically for radar data, including: a basic simple extrapolation through an advection field, the Spectral Prognosis (S-PROG) model and the Short-Term Ensemble Prediction System or STEPS nowcast model [9,10]. These pySTEPS nowcast models have been used in many studies around the world to evaluate their forecasting skill and to compare them with other types of nowcast models, using radar

data [8,11–17] and other types of data sources such as microwave links [18], and blended NWP data [19].

On the other hand, satellite data have not been widely used as input. Several nowcast models have been developed to use satellite data to improve or extend radar data nowcast models (e.g., [20–22]), while other models have been designed to operate primarily with satellite data (e.g., [23–28]). These nowcast models often utilize threshold values of satellite brightness temperatures or reflectance values in specific wavelengths, to determine areas of convection and water vapor concentration prior to or at the early stages of precipitation events. These detected areas have been used to track any convective development, and in some cases, to assign probabilities of convective storms [23]. Other studies have used satellite-derived precipitation data as inputs in different types of nowcast models (e.g., [29–32]).

One variable that has been observed to be a possible indicator of future convective development has been the Split-Window Difference or SWD [33–35]. The SWD is defined as the difference in the brightness temperature measured at the 10.33 µm wavelength (called clean-window) and at the 12.3 µm wavelength (called the dirty-window). The difference in these two spectral channels describes the amount of low-level water vapor in the atmosphere because water vapor absorbs and re-emits large amounts of energy in the dirty-channel. On other hand, the clean-window is transparent and is not affected by the water vapor absorption, and all the energy comes from the surface of the earth (which is warmer than the clouds). In other words, the difference measured between these two wavelengths is related to the amount of water vapor in the atmosphere. The SWD is small when there is water vapor in the lower atmosphere and the difference between the dirty and clean window is small. The SWD has higher values when the water vapor concentration is low. Although there are other variables that can affect the accuracy of the low-level water vapor measurements obtained through the SWD [36], this study uses the simple form of the SWD. This is because the region does not have other instruments available, such as ground base weather stations, which would cover the whole area of study and provide the necessary variables to correct the SWD measurements (such as surface temperature).

The first objective of this study was to evaluate the SWD as an indicator of rainfall and convective development, using GOES-R and radar data across central Mexico. The second and main objective was to utilize the SWD and radar rain rate data to assess the performance of two nowcast models provided by the pySTEPS library to evaluate their performance within the region. The premise behind using the SWD was to generate a forecast, using the SWD signal as an indicator of convective activity prior to the onset of rainfall.

## 2. Materials and Methods

The data used in this study came from two sources. The first was from NOAA's GOES-16 satellite, using the CONUS sector, and the second was from the Queretaro weather radar. The analysis period spanned 47 days, from 1 July 2018 to 16 August 2018, representing the time when the radar dataset was most complete during the rainy season.

The SWD was derived using channel 13 (10.3 µm band) for the "clean-window" and channel 15 (12.3 µm band) for the "dirty-window", as described by NOAA [37,38]. The SWD was calculated using SWD = T(B, Dirty) − T(B, Clean), where T(B,) represents the brightness temperature. These two channels have a spatial resolution of 2 km, and a temporal resolution of 5 min.

The Queretaro weather radar is a Doppler, "C" band radar with pulse compression [39], located at $20.7802°$ Lat, $−100.5504°$ Lon, approximately 27 km northeast of the city of Santiago de Queretaro, Queretaro, Mexico. It operates within a range of 239.75 km, with a beam width of $1°$ and a temporal resolution of 5 min. The radar data underwent a quality control procedure using the Gabella and PIA methodology [40,41], respectively. The rain rate was calculated using the Z-R relation with the parameters a = 220, and b = 1.6. After performing a visual quality control of the radar data, 5 days were excluded from the radar data set: 2 July, 17:42 Z, to 5 July, 17:42 Z; 16 July, 2:57 Z, to 17 July, 14:07 Z; and 16 August

starting at 2:02 Z. The prevalent type of precipitation during this time of year in the central Mexico region is convective precipitation.

The methodology utilized to evaluate the SWD as an indicator of convective and rainfall development was to analyze the relationship between this variable and the radar rain rate. To achieve this, a common grid was required. For this study, the satellite's 2 km by 2 km grid was chosen as the common grid, illustrated in Figure 1, due to its compatibility with pySTEPS. The radar data were interpolated into the satellite grid using the nearest-neighbor interpolation.

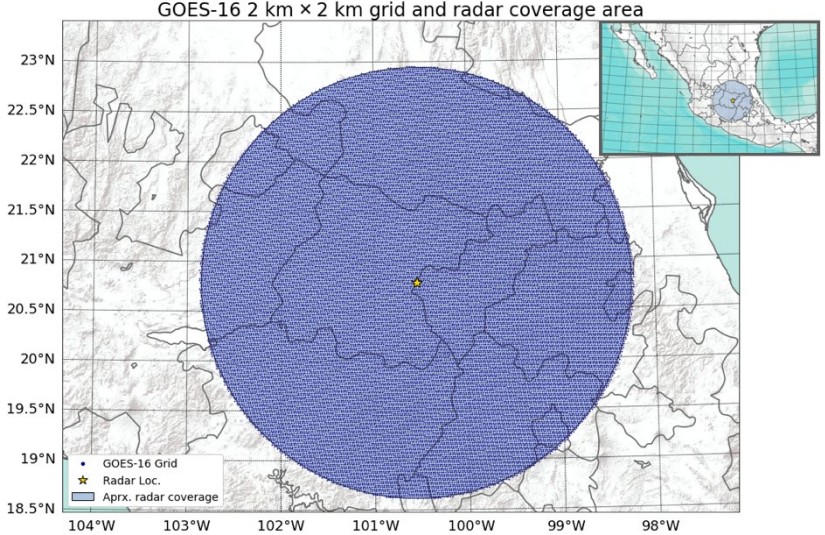

**Figure 1.** GOES-16 satellite 2 km by 2 km grid used as common grid over the Queretaro radar coverage area for rain characterization. Basemap sources: Esri, USGS, NOAA [42].

For the assessment of SWD nowcast, we employed the nowcasts provided by the pyS-TEPS library of Python [9,10]. Although the pySTEPS (v. 1.4.1) library includes five types of nowcasts, only two were used for this study: extrapolation and S-PROG. Extrapolation is the simplest nowcast method available in pySTEPS, which is an extrapolation method based on advection. Essentially, the model assumes a constant intensity of precipitation and moves the provided data through a given advection field. On the other hand, the S-PROG nowcast, as described by [1], is an extrapolation through advection nowcast method that incorporates spectral decomposition in the spatial field, integrating an autoregressive model within each cascade field. These two nowcast methods were chosen due to their efficiency in computational resources, allowing forecasts to be obtained in less than 1 min. Furthermore, given that the nowcasts were intended for use with a satellite variable rather than radar data, understanding how they would impact the intensity of the input variable was crucial. Both Extrapolation and SPROG models modify the intensity of the input variable; however, the changes are a consequence of how the models operate and not a direct modification of the intensity by the algorithm. In other words, the algorithms of Extrapolation and S-PROG do not involve the direct modification of the intensities, such as including dissipation factors or adding stochastic variations to the input fields [9].

Although all motion fields calculation functions through pySTEPS were tested, the Lucas–Kanade method (LK) pySTEPS function with the default parameters was used to derive the motion fields for all nowcast methods because its minimal computational requirements and its ability to provide a reliable representation of the motion field. The evaluation of these methods is based on the dichotomous verification statistics provided in the pySTEPS library, which include the Probability of Detection (PoD):

$$PoD = \frac{hits}{hits + misses} \tag{1}$$

False Alarm Ratio (FAR):

$$FAR = \frac{falseAlarms}{falseAlarms + hits} \tag{2}$$

and the Heidke Skill Score (HSS):

$$HSS = 2\frac{hits \times c.negs - f.a. \times misses}{(hits + misses) \times (misses + c.negs) + (hits + f.a.) \times (f.a. + c.negs)} \tag{3}$$

In addition to using pySTEPS's nowcast models, the Thunderstorm Detection and Tracking (DATing) module was also used to obtain SWD clusters that could lead the identification of possible convective development. The DATing module uses the TRT Thunderstorms Radar Tracking algorithm [2,9] developed by MeteoSwiss. This algorithm involves the initial detection of storm clusters, and then it uses pySTEPS advection functions to generate motion fields to obtain the paths of the detected storm clusters over a specific time period. The DATing module was used in this study to filter out individual pixels or small groups of SWD data, assuming that these will most likely not lead to strong multi-cell convective storms. The DATing module uses natively the radar reflectivity as input, for the SWD data; the tracking parameters were modified to improve the track, and the detailed process are in Section 3.2.1.

Finally, the evaluation was performed for a period between 1 h and 35 min before the onset of a precipitation event and 4 h and 55 min after the start of a precipitation event. This time frame was used in the DATing module to obtain the SWD clusters for each event. Each image or data matrix available in this 6:30 h interval (an image every 5 min), for all precipitation events occurred during the 47-day analysis, was used as input for the nowcasts. Figure 2 illustrates the study area for the nowcast evaluation, which is the 2 km by 2 km satellite/common grid of Figure 1 beyond the radar range.

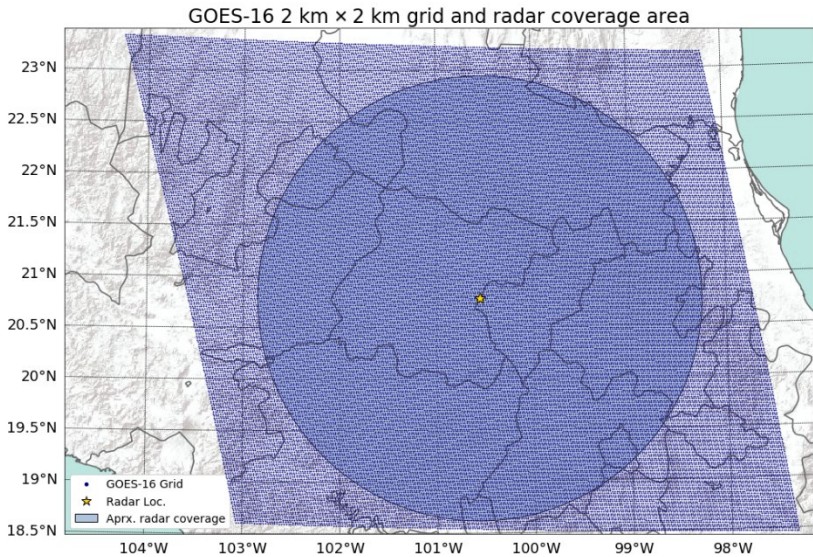

**Figure 2.** GOES-16 satellite 2 km by 2 km grid used as common grid over the Queretaro radar coverage area for the evaluation of the pySTEPS nowcast models. Basemap sources: Esri, USGS, NOAA [42].

## 3. Results

### 3.1. SWD Relationship with Radar Rain Rate

In this part of the study, each pixel in the common grid was considered a case, using pixel-by-pixel comparison. Figure 3 illustrates the relationship between SWD and radar rain rate, revealing that the majority of rain rate cases shows a positive SWD. Extreme rain rates (100 mm/h or above) were associated with SWD values ranging from 0 K to 5 K, with an average of 1.99 K. Figure 4 displays the distributions of the SWD for pixels with rain (left

graph) and pixels without rain (right graph). The left graph of Figure 4 suggests that most cases where precipitation occurrence had an SWD between 0 K and 5 K, with the highest percentage of cases (13.5%) falling within the 1.23 K and 1.44 K interval. Conversely, for the non-rain cases displayed in Figure 4b, the majority of cases exhibited SWD values between 0 K and 10 K, with a significant concentration between 5 and 10 K. The highest percentage of cases (4.1%) occurred within the SWD range of 1.86 K to 2.07 K. The main differences observed between the no rain cases distribution and the rain cases distribution was the number of cases above and below 2.5 K.

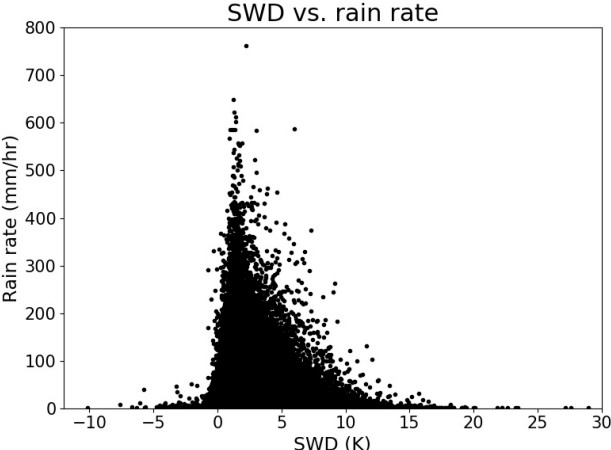

**Figure 3.** Scatter plot of SWD vs. radar rain rate for all cases analyzed.

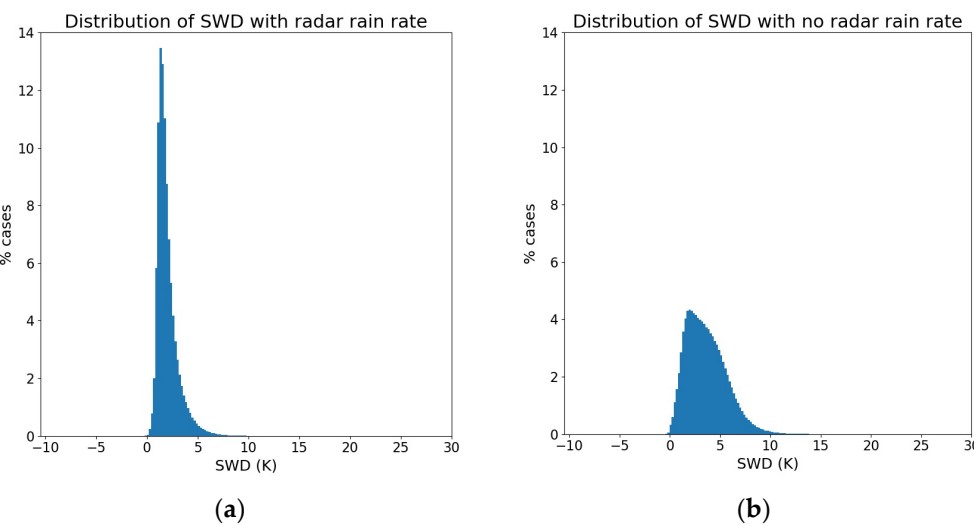

**Figure 4.** Distributions of (**a**) SWD cases with radar rain rate and (**b**) SWD cases without radar rain (right). Non-rain cases are SWD cases with rain rate less than 0.5 mm/h.

A notable distinction between the distributions of no rain cases and rain cases was the frequency of occurrences above and below 2.5 K. For the distribution of cases without rainfall, 37.2% exhibited SWD values lower than 2.5 K, whereas for the distribution of rain cases, 82.2% of cases had SWD values below 2.5 K, suggesting that SWD values below 2.5 K were commonly associated with precipitation.

While Figures 3 and 4 suggest a weak correlation between radar rain rate and SWD, an analysis of SWD values prior to the onset of rainfall was conducted to gain insights into the behavior of SWD during convective development. Figure 5 shows the average SWD at different offset times for various rain rate intervals. These offset times represent the time between the start of precipitation and the satellite data collection, indicating that the average SWD was measured at the specified offset time before the recording of rainfall.

Across all cases, the average SWD has an increasing trend with offset time, indicating that the average SWD decreased as the time drew closer to the onset of precipitation events. The average change between the 0- and 240-min offset times was 1.89 K; and between 0- and 90-min offset times was 0.81 K. This rate increased with the rain rate intensity, with the most significant change observed for the rain rate interval of >200 mm/h (3.13 K) for 0- to 240-min offsets, and 2.53 K for the 0- to 90-min offset. Furthermore, this graph also shows that the average SWD decreased as the rain rate increased for offset times lower than 20 min. Notably, for extreme events, the SWD exhibited a 2 K decrease one hour before the onset of rainfall. This decline in SWD preceding precipitation events was also observed by [33], where a storm case study analyzed the SWD derived from the GOES-16 data, revealing that the SWD dropped below 3 degrees C just before and during the onset of rainfall. Figure 6 shows the change in average SWD per 5 min for all offset times. In general, the change was negative, indicating that the SWD decreased before the development of rainfall. The most substantial decrease was observed for high intensity precipitation between the 5- and 90-min offset time. Finally, the values also formed a local minimum or valley between the 75- and 25-min offset times, with the maximum change occurring between 50 and 45 offset minutes. The average change that occurred during this downhill was of 0.6 K per 5 min for all cases.

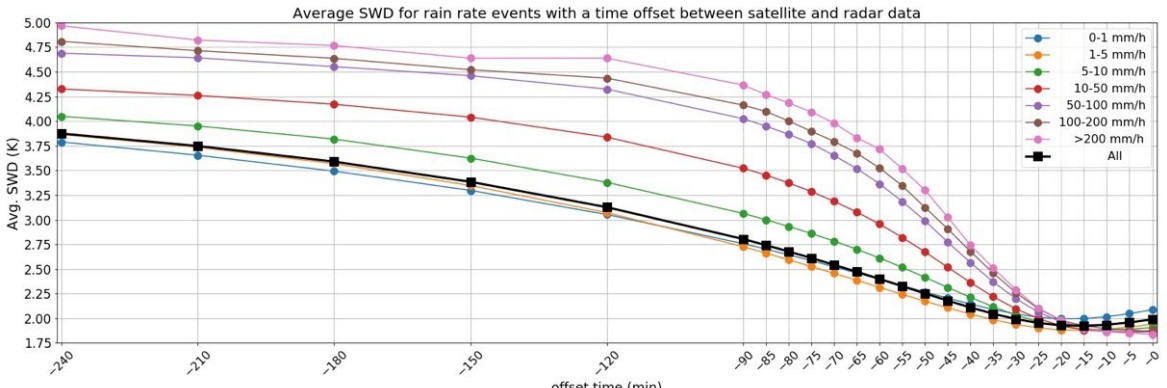

**Figure 5.** Graphs of the average SWD for each offset time (offset between the radar data and the SWD data) for all cases and for the rain rate intensity intervals indicated. Negative times on the *x*-axis indicate the time before the measurement of rain rate (0 min is time of measurement of rain rate).

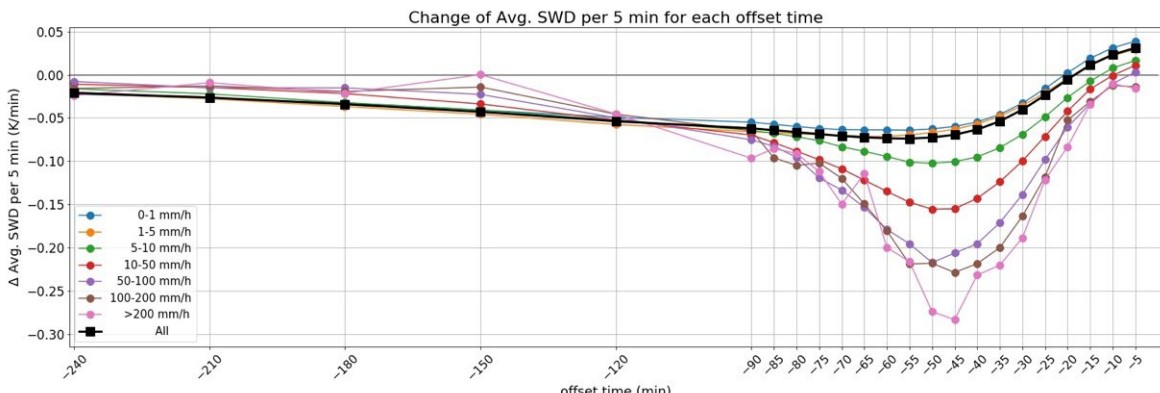

**Figure 6.** Graphs of the change in the average SWD per 5-min for each offset time (offset between the radar data and the SWD data) for all cases and for the rain rate intensity intervals indicated. Negative times on the *x*-axis indicate the time before the measurement of rain rate (0 min is time of measurement of rain rate).

The results demonstrate the feasibility of setting a threshold SWD value in order to identify rainfall development areas. Using Figure 5, this threshold value was determined

to be 2.5 K, since the average SWD drops below this value approximately 60 min prior to the measurement of rain rate. Thus, if any significant SWD areas fall below 2.5 K, there is a strong likelihood of rainfall development within the following 60 min.

Diurnal Cycle

In addition, a SWD diurnal cycle was derived for the rain rate and the SWD. This was performed by calculating the average of radar rain rate and SWD over the entire common grid for each image available in the analysis, observing how it varied throughout the day. Moreover, the radar rain rate average (RR_aa) was calculated by replacing NaN values with 0 to have a consistent methodology to compare with the SWD area averages (SWD_aa) values.

Figure 7 shows the running mean for all available days for the RR_aa and the SWD_aa, as well as the hourly average of both. The mean values show that rainfall typically commenced after 11 h. local time, with a maximum around 17 h. local time. On the other hand, the SWD_aa tended to increase at around 6 h. local time and its maximum occurs at 12 h. local time. The behavior of RR_aa is consistent with the observed diurnal cycle of precipitation in various regions during the rainy season [43–45]. The graphs also highlight the inverse relationship between SWD_aa and the presence of precipitation, with a noticeable decrease in SWD_aa coinciding with an increase in RR_aa. Additionally, the rise in SWD observed around 6:00 a.m. local time corresponds to the sunrise, indicating a decrease in low-level water vapor levels with the presence of sunlight. It is important to emphasize that both SWD_aa and RR_aa were averages over the entire region, considering all areas where there is no rain rate as 0.0 mm/h. Consequently, the observed maximum and minimum SWD_aa values exceeded those reported in the previous section.

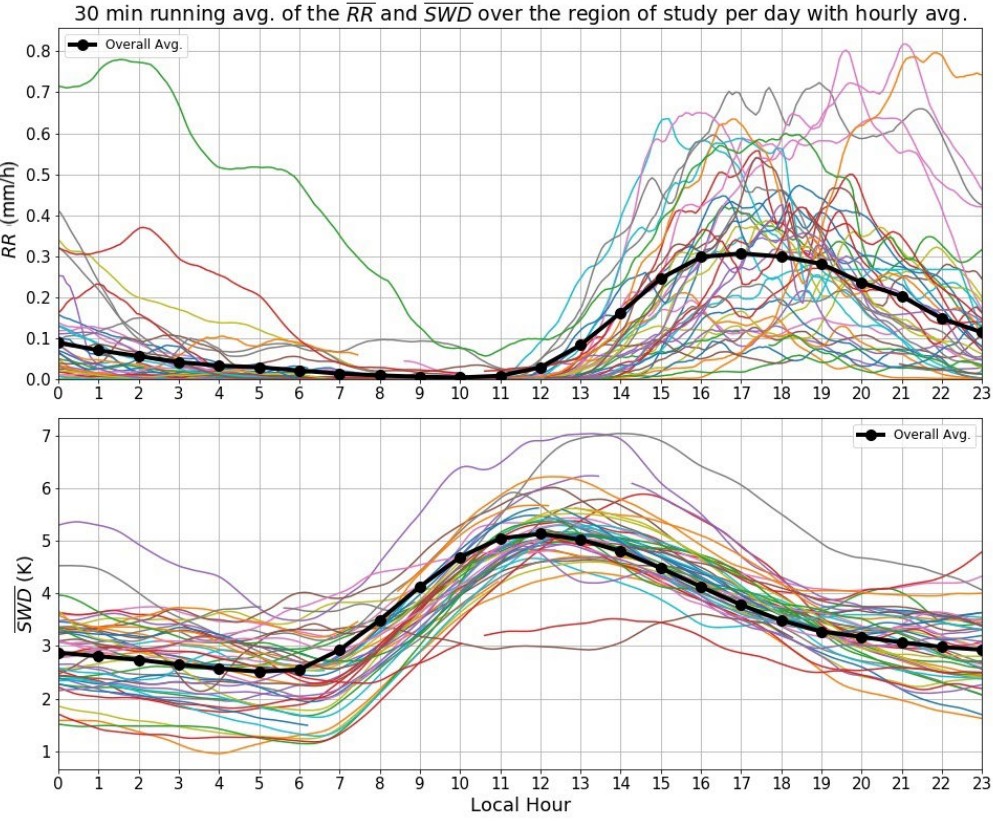

**Figure 7.** The 30-min running mean of the area average of the radar rain rate (marked with a bar over RR) and SWD (marked with a bar over SWD) per day for available days with the hourly average (black line).

### 3.2. Results of the Nowcast Model Evaluation

### 3.2.1. Why SWD?

Although the previous section reveals an unclear relationship between rain rate intensity and SWD, a change in the SWD was observed between 90 and 60 min prior to precipitation occurring. In other words, the decrease in the SWD below 2.5 K indicates the likelihood of rainfall within 60 min for the region. Figure 8 shows the SWD field, for values between 0 and 2.5 K and radar rain rate of one of the events analyzed, 8:07 Z or 3:07 h local time, 9 August 2018. This figure highlights how the SWD values can indicate potential areas of convective development one hour before and after the rain event started. However, this example also shows that not all SWD areas below 2.5 K developed into precipitation, suggesting that while the SWD can overestimate potential precipitation areas, it remains a valuable tool for identifying potential convective development ahead of rainfall. Similar instances, as depicted in Figure 8, can be found in the Supplementary Materials (Figure S1). These observations align with findings from previous studies utilizing satellite data, including GOES-16 data [33–35]. The SWD data can be utilized as input in nowcast models to predict potential precipitation areas before the initial radar measurements of rainfall. Once the precipitation event commences, radar data become more reliable for predicting rain intensity, while the SWD can aid in indicating the possible direction of precipitation alongside radar data, though not its intensity due to the lack of a clear relationship between rain intensity and SWD. Ultimately, the combined use of SWD with nowcast models facilitates short-term predictions of potential precipitation areas before the first radar measurements, followed by the use of radar observations to forecast storm behavior.

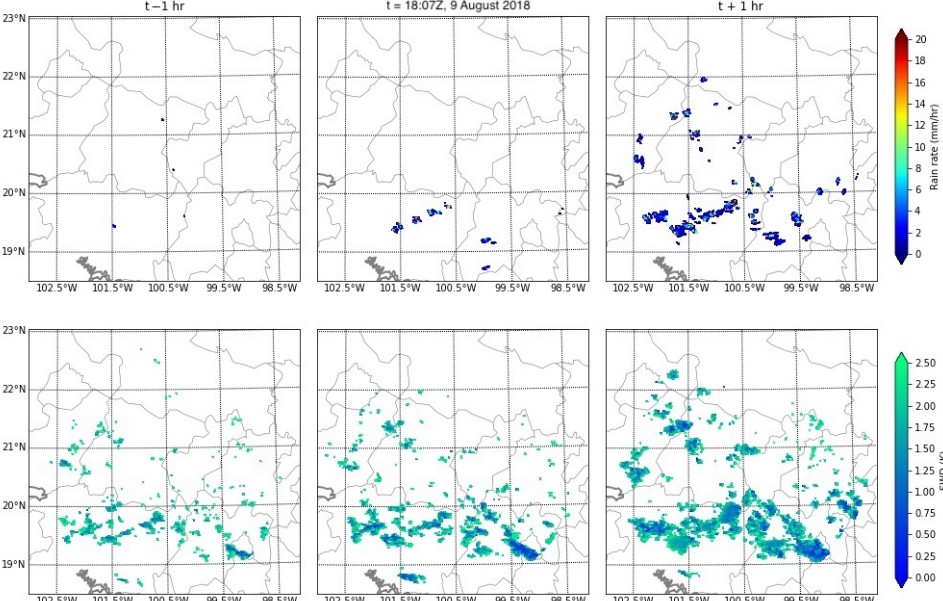

**Figure 8.** Rain rate from the Queretaro radar (top images) and SWD field below 2.5 K from the GOES-16 satellite (bottom images) for the event starting at 18:07 Z, 9 August 2018. Left images are for one hour before start of event, middle images are of start of event and right images are of 1 h after start of event.

### 3.2.2. Setup SWD and Event Selection

The SWD field requires modification, in order to work properly in pySTEPS, because the algorithms implemented in pySTEPS worked optimally when the input data matrix contains NaN flags for missing values, clusters or groups of data points. Therefore, the SWD field had to be masked using the 2.5 K (negative values were eliminated) threshold to select only areas where rainfall is plausible. Additionally, the values had to be multiplied by 100 to amplify the changes in the field. With the filtered SWD field, the nowcast models and the LK function

to obtain the motion field were used and the model ran without any issue. Henceforth, any reference to "all SWD field" or "all of the SWD field" or "the SWD field" refers to this filtered SWD field, unless specifically stated otherwise. The following DATing tracking parameters were observed to yield the best results for the SWD data and were consequently used for this study: (units × K × 100) *minref* (lower threshold value) = 100, *maxref* (maximum threshold value) = 250, *mindiff* (minimum difference between two maxima in one area needed to distinguish objects) = 60, *minsize* (minimum size of cluster) =10 (pixels), *minmax* (minimum of maximum values) = 240 and *mindis* (minimum distance between clusters) = 5 (pixels).

The second obstacle to tackle prior to evaluating the nowcast models was determining the events to be analyzed. The events were selected using a 30-min running mean over the radar rain rate, presented in Figure 9. The start of each event was defined as the occurrence of a local minimum in the data after 10 h local time and before 20 h local time, a period with high density of rain events. To select the events efficiently and computationally fast, the second derivative test was used to identify the local minimum. After locating the local minimum, the start of the event was selected as the time when the average precipitation reaches the value of 0.002 mm/h after the local minimum. This was performed in order to eliminate any potential noise sources during the local minimum calculation.

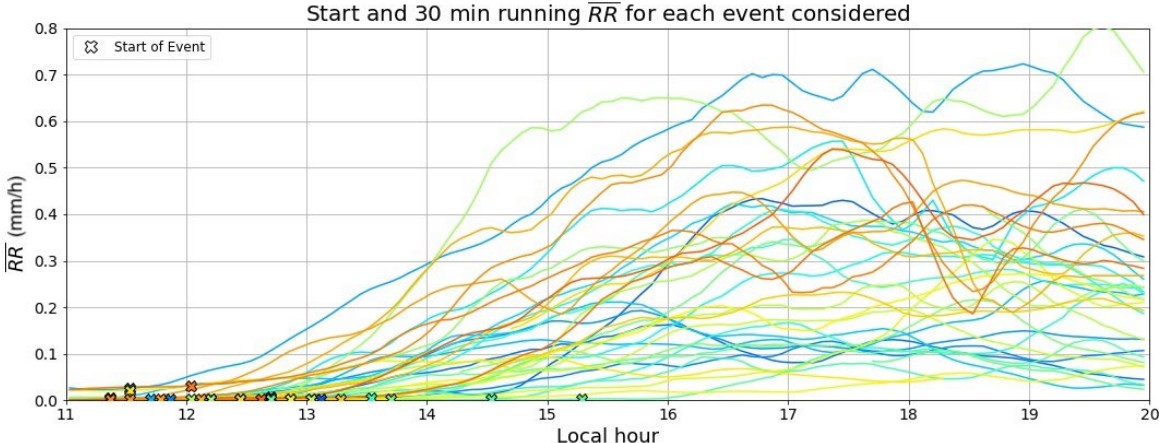

**Figure 9.** The 30-min running mean of the area average of the radar rain rate (marked with bar over RR) of the events considered for the nowcast evaluation with the start of each event.

From this methodology, 36 events were selected (23 for July and 13 for August). This method can be considered efficient because, as Figure 6 suggests, assuming one event per day, the process identified 36 of 43 possible events, resulting in an 83% efficiency rate.

3.2.3. Evaluation of Nowcast Models Using SWD Data

This section presents the statistics comparing all SWD fields as input and those obtained only using the SWD clusters as input in the models. The forecast time was set to 15 min, because it was the shortest useful lead time for both the SWD and the rain rate. In addition, shorter lead times result in more accurate forecasts, and, therefore, the results from the 15-min forecast were considered the most accurate useful predictions that can be obtained in this analysis. The input data for the models and the resulting forecast fields were on a logarithmic scale, but the evaluation of the forecast was performed by transforming the logarithms into K and mm/h. Furthermore, for consistency, any values of 0 K given by the models were converted to NaN.

Figure 10 shows the scatter plots of the statistics used to evaluate the two nowcast models, comparing the results between the SWD field (All SWD field) and the SWD clusters (SWD clusters). The scatter plots indicate that SWD clusters provided better overall results. The PoD, FAR and HSS results show that the Extrapolation and S-PROG models were accurate in general, with approximately 93% of cases with PoD of 0.7 or higher, 71% of the cases had a FAR below 0.3, and 68% of the cases had a HSS at or above 0.7. Figure 11

presents all the average statistics in every 5-min interval of the analysis period and the difference between the average results obtained using the SWD field and SWD clusters. The graphs illustrate that the results improved as the time elapsed. Finally, the SWD clusters gave more accurate forecasts throughout the whole period of analysis. Although several nowcast models use satellite data [23–32], there have been no prior studies employing SWD satellite data as inputs in the pySTEPS nowcast models. While some studies have utilized the optical flow fields available in pySTEPS with satellite data [46] and not the nowcast models themselves, certain studies have evaluated S-PROG and Extrapolation pySTEPS nowcast models with radar data, using some of the same dichotomous statistics. The authors of [8,11,12] conducted evaluations of the two models in the USA (former two) and China, respectively, with radar data for precipitation events. Their PoD and FAR results closely resemble those obtained with the SWD data, with values between 0.6 and 0.75 for average PoD and values between 0.2 and 0.4 for average FAR for lead times near 15 min. However, because [8,11,12] involved radar data rather than satellite data, caution should be exercised when comparing these results with those obtained using the SWD.

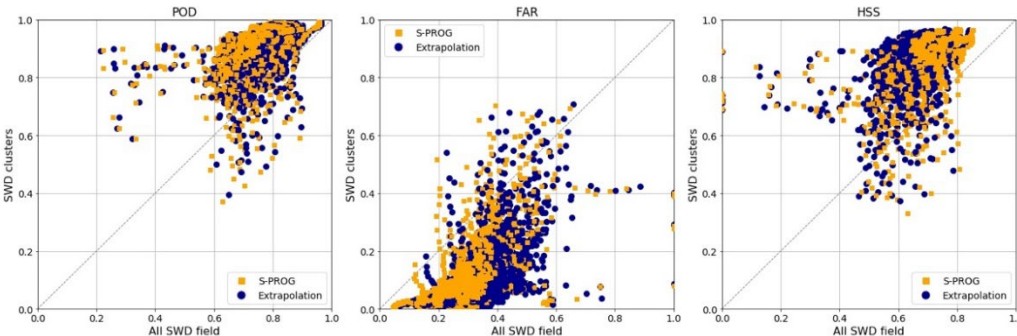

**Figure 10.** Scatter plots of all the evaluating statistics obtained using all the SWD data per data matrix used vs. using only the SWD clusters detected and tracked by the DATing module for the 15-min forecast.

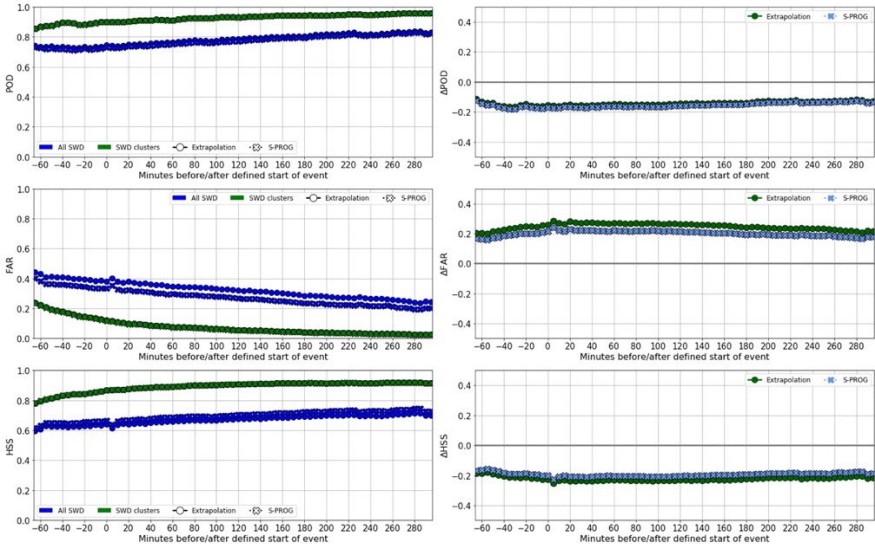

**Figure 11.** Graph of the average statistics indicated (POD, FAR, HSS) of the 15-min forecast, using the image of the time indicated as input (graphs on right column) and the difference between "All SWD" and "SWD clusters" (left column, marked with a Δ). The time is every 5-min interval before (marked with a negative sign) and after the defined start of the event for SWD data.

Although Figures 10 and 11 give an idea of the nowcast behavior for each type of SWD input, it is helpful to have a visual representation of the forecast to understand the previous results. Figure 12 shows an example of the 15 min forecast for an event that started on 1 July 2018, 18:57 Z, or 13:57 h local time. The images show that both models provided

similar forecasts. The forecast using SWD clusters was more accurate. The SWD clusters obtained from the DATing module filtered out a lot of smaller SWD areas; therefore, the errors that arose from not predicting the movement of these small SWD clusters correctly were not considered. These errors, although small in magnitude, were significant in quantity, providing an explanation as to why utilizing the entire SWD field (Figure 12) led to less accurate overall forecasts. However, it is important to note that the DATing algorithm does not always detect all large clusters, potentially resulting in missed areas of precipitation.

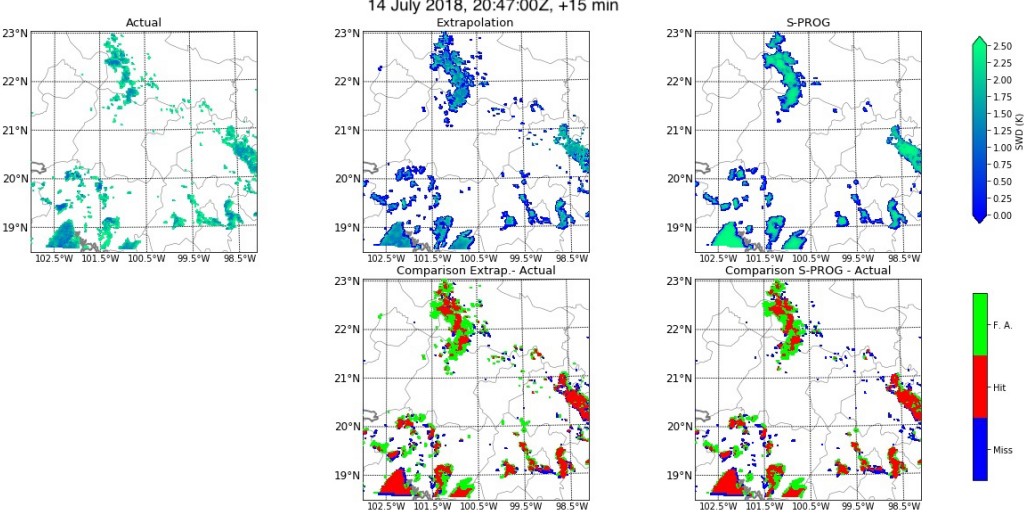

**Figure 12.** Images of the SWD field for the actual data and the 15-min forecast obtained using the two nowcast models indicated (**top**) occurring on 14 July 2018, 20:47:00 Z; and comparison between the actual data and the forecast data (**bottom**).

### 3.2.4. Evaluation of Nowcast Models Using Radar Data

The same evaluation of the Nowcasts models presented in Section 3.2.3 was also conducted using the Queretaro radar data. To preserve the consistency between the SWD and the radar data, the results are also presented as a comparison between the results obtained using the rain rate field (all_RR) and the rain rate that fell into a SWD clusters (RR_in_SWD_clusters). Lastly, the threshold value used to separate rain and no rain was set as 0.5 mm/h (since values below this threshold were considered noise in the original rain rate data).

Figure 13 shows the scatter plots results for the two models; the results for the PoD and HSS show that using all_RR was less accurate than using RR_in_SWD_clusters. For both models, the PoD of 93% of cases fell below 0.7 and the PoD of 58% of cases fell between 0.7 and 0.5. For the FAR, 87% of the cases were below 0.3. The difference between using as input all_RR and RR_in_SWD_clusters was negligible. Figure 14 shows statistics results for a 15-min forecast, averaged for every 5 min interval after the start of the rain event. These graphs show that, on average, using all_RR field resulted in similar forecast as using RR_in_SWD_clusters. Additionally, the PoD tended to increase on average by almost 0.4 in the 295 min after the start of the event, the FAR decreased by 0.2 in the same period. These results for all_RR are not as accurate as those obtained by previous works. The authors of [8] evaluated the models for 10 events in the Texas, USA region and obtained PoD values of 0.6 and 0.75 for Extrapolation and S-PROG, respectively, and FAR values between 0.3 and 0.4 for both models. The authors of [11] used 80 events from Texas and Colorado, USA, and radar reflectivity with S-PROG and Extrapolation, which resulted in average PoD of approximately 0.75 and average FAR of 0.2 for the 15-min lead time. Lastly, the authors of [12] used the Extrapolation and S-PROG models with radar reflectivity data from China for over 100 events. They found that for short lead times (12 and 24 min), the POD reached 0.75 and the FAR was below 0.2. However, the PoD values were

closer to those observed by [8,11,12] as the events developed. Finally, an example of these forecasts (Figure 15) showed similar observations as those made with the SWD data for the same event, revealing comparable results between the Extrapolation and S-PROG models. Additionally, both models tended to overestimate the area affected by rainfall.

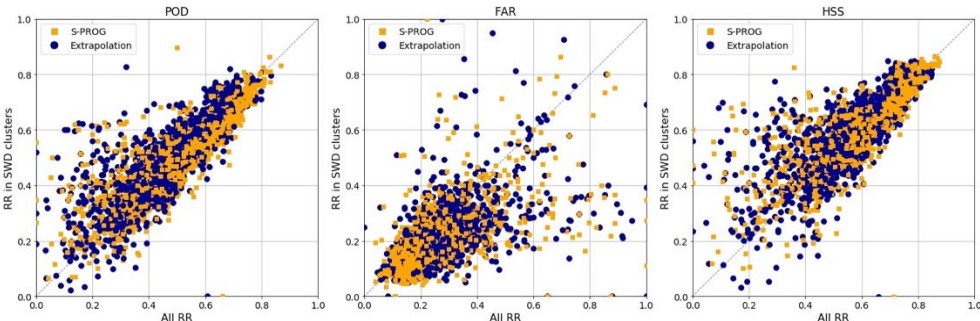

**Figure 13.** Scatter plots of all statistics obtained using all the rain rate data per data matrix used vs. using only the rain rate in SWD clusters detected and tracked by the DATing module for the 15-min forecast.

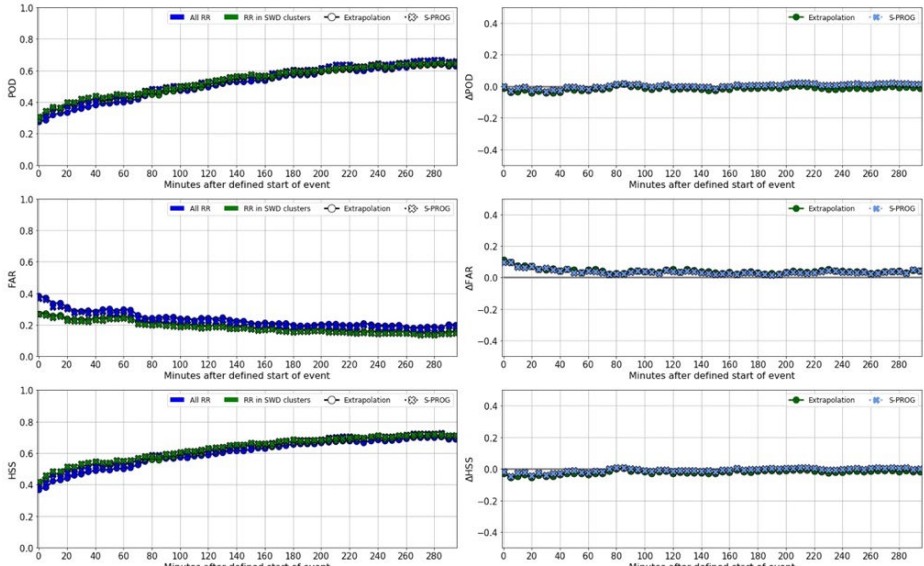

**Figure 14.** Graph of the average statistics indicated (POD, FAR, HSS) of the 15-min forecast, using the image of the time indicated as input and the difference between "All RR" and "RR in SWD clusters" (left column, marked with a Δ). The time is every 5-min interval before (marked with a negative sign) and after the defined start of the event for SWD data (left).

As previously mentioned, the results obtained using radar data were generally less accurate compared to those derived from SWD data. This disparity can be attributed to the larger volume of SWD data and the comparatively larger SWD clusters, which increased the likelihood of hits. However, the FAR results were marginally better for the radar data. This inconsistency can be traced back to how the nowcast processed the SWD and radar data, as depicted in Figures 12 and 15. Both nowcast models tended to generate false alarms and misses along the edges of the clusters. Nonetheless, the models produced a greater number of false alarms for SWD than for rain rate, likely due to the models not being designed specifically for forecasting satellite products. The discrepancy in the number of false alarms created by the models for each type of data influenced the statistics that relied on false alarms, such as FAR. Consequently, the average FAR for the SWD was approximately 0.1 higher than that for the radar data.

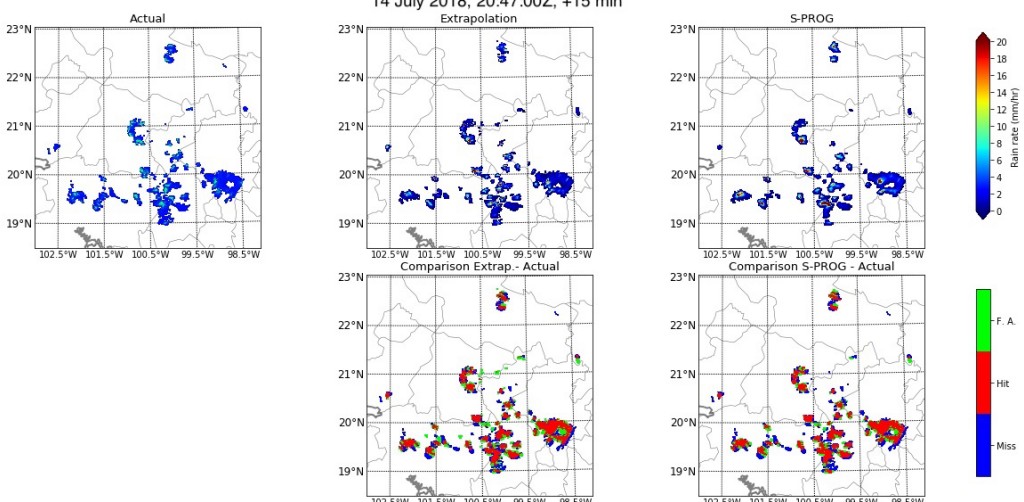

**Figure 15.** Images of the RR field for the actual data and the 15-min forecast obtained using the two nowcast models indicated (**top**) occurring on 14 July 2018, 20:47:00 Z; and comparison between the actual data and the forecast data (**bottom**).

### 3.2.5. Evaluation of Nowcast Models with Variation in Coverage Area

Previous results showed that the nowcasts gave less accurate results when radar data were used as input. This might be due because the radar has some limitations in the coverage area. To explore if this limitation had any effect in the results, the nowcast models were evaluated by applying two radar coverage area limits. The first constraint was the whole radar coverage area (all radar area; Figure 1); and the second was a constrain the radar range to 180 km (180 km radar area). Figure 16 shows the differences in the average PoD and FAR between all radar area, and 180 km radar area. The 180 km radar area demonstrated improved results for both PoD and FAR during the initial 60 min following the event's commencement. However, this improvement diminished to less than 0.05 for the subsequent period.

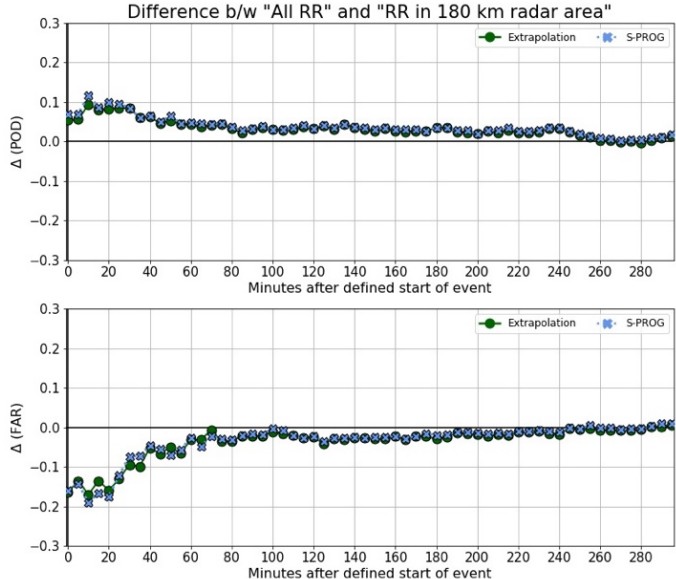

**Figure 16.** Graph of the differences in (**top**) average POD and (**bottom**) average FAR for the 15-min forecast, averaged every 5-min, the difference is between all radar area and the 180 km radar area.

### 3.2.6. Real-Data Pixels and Statistics Results Relationship

Following the findings of the previous section, which indicated a change in nowcast skill corresponding to decreasing coverage, a question emerged regarding the significance of NaNs in the analysis. Consequently, the subsequent analysis focused on assessing the impact of NaNs on the applied statistics. One of the primary motivations for this investigation was the observation that the radar data contained fewer data values within each image than the SWD data, resulting in outliers exerting a substantial influence on the outcomes. To test this hypothesis, a comparison was made using the fraction of pixels containing real data (non-NaN) for each variable across all cases, aiming to identify any potential correlations with the results.

Figure 17 shows the scatter plot of the logarithm-10 of the fraction of pixels in each image with real data (pixel fraction) and the PoD for each nowcast model divided into SWD and radar data, as well as their linear correlation for the two nowcast models. All data sets had a positive, linear correlation with the log-10 of pixel fraction. For the SWD data, the data were more concentrated in the PoD values above 0.5, but the outliers present were very separated from the main cluster of data. For the radar data, the data remained below 0.8 and were more scattered, unlike the SWD, which did not have many outliers very separated from the main data cluster. Figure 18 shows the same but for FAR. This variable has a negative, linear correlation with the log-10 of pixel fraction. The SWD data were less spread with some outliers, and the radar data were more spread, especially for pixel fraction values below 0.003 (approximately 186 pixels). The linear correlation observed in the graphs was calculated using the Pearson's correlation (R) and Spearman's Rank (rho) (Table 1). This table shows R and rho values for each of the statistics studied and the log-10 of the pixel fraction. All variables showed a high correlation with pixel fraction, except for the HSS for SWD, which had a moderate correlation.

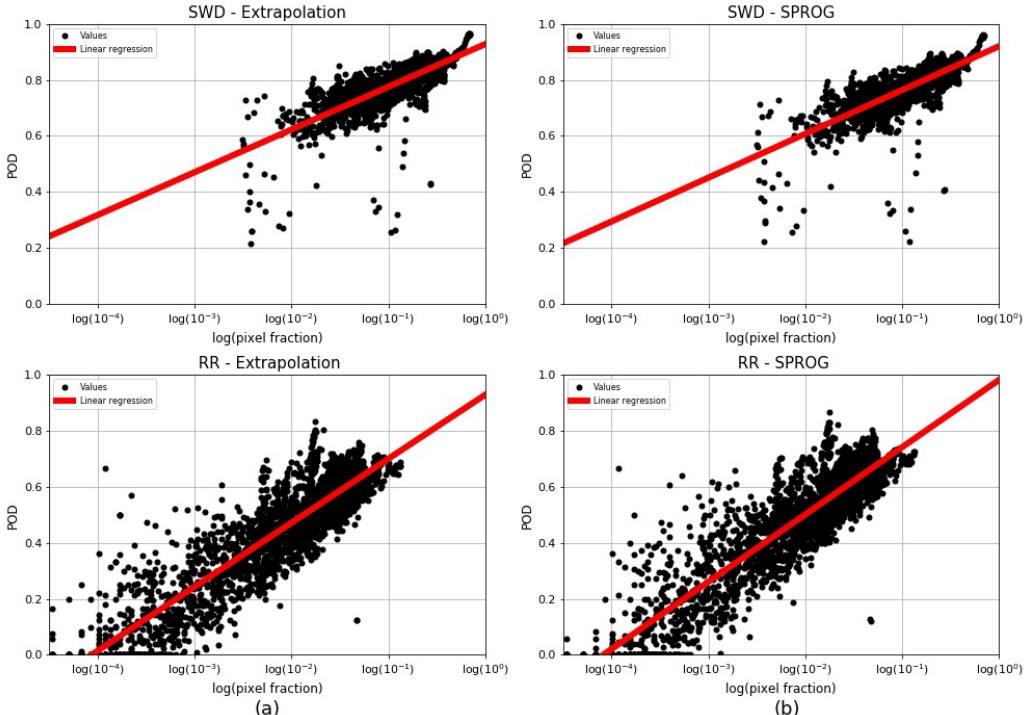

**Figure 17.** Scatter plot of the log-10 of the pixel fraction (real data pixels over total pixels) and the PoD for the events analyzed. Bottom graphs are of radar rain rate and top graphs are for SWD results for (**a**) Extrapolation and (**b**) S-PROG.

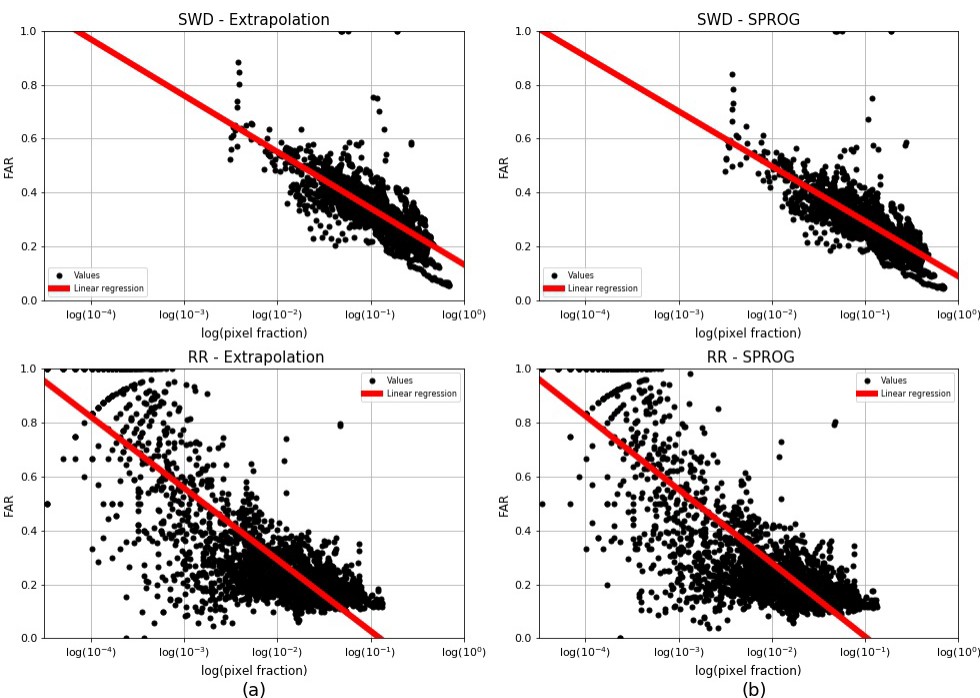

**Figure 18.** Scatter plot of the logarithm-10 of the pixel fraction (real data pixels over total pixels) and the FAR for all cases and events of radar rain rate and SWD results for (**a**) Extrapolation and (**b**) S-PROG.

**Table 1.** Results of the correlation coefficients between the fraction of total pixels with real data and the statistics results for all nowcast models. Asterisk (*) marks the use of the logarithm-10 of the pixel fraction (real data pixels over total pixels) for the correlation analysis.

| Statistic | Model | Pearson's R | | Spearman's Rank | |
|---|---|---|---|---|---|
| | | **SWD** | **RR** | **SWD** | **RR** |
| PoD * | Extrap | 0.77 | 0.91 | 0.83 | 0.88 |
| | S-PROG | 0.78 | 0.91 | 0.84 | 0.87 |
| FAR * | Extrap | −0.79 | −0.84 | −0.83 | −0.73 |
| | S-PROG | −0.79 | −0.84 | −0.84 | −0.73 |
| HSS * | Extrap | 0.55 | 0.91 | 0.60 | 0.87 |
| | S-PROG | 0.60 | 0.91 | 0.66 | 0.87 |

The results of Figures 17 and 18 and Table 1 clearly indicate a strong correlation between the nowcast evaluation statics and the quantity of real data (non-NaN) pixels within an image. The nowcast evaluations demonstrate superior results with a higher number of real data pixels present in the input image. Notwithstanding, an important point from the results needs to be addressed: The difference between the relationship of the variables and the pixel fraction when using the SWD and using the radar data. The difference can be observed in the variation in the R and rho values of Table 1 for each type of data, and more clearly in the plots of Figure 18. The discrepancy arises from the nowcast's tendency to generate a greater number of false alarms when utilizing the SWD compared to radar data, primarily because the nowcast models are not specifically designed to predict the SWD. Consequently, this discrepancy significantly impacts the statistics reliant on false alarms when utilizing SWD data.

### 3.2.7. Deeper Evaluation of Nowcast Models with Radar Data

In general, the results using radar rain rate data were less favorable than those obtained using SWD data, particularly during the initial stages of precipitation events. Although

Section 3.2.6 demonstrated the influence of the real data points in the input image on the results, there exist other potential factors that could affect the nowcast skill outcomes. One of these factors is the radar's ability to "see" or accurately measure the entire region covered by the radar range.

An initial assessment was to observe how the nowcast skill varied with radar range. This was performed by dividing the radar coverage area into rings, 10 km in width, and obtaining the evaluation statistics for each ring. Figure 19 shows the average PoD and FAR per outer ring radius. For PoD, the average value was around 0.2 near the radar, and increased gradually until reaching 40 km, when the PoD decreased to slightly less than the initial value and continued with a gradual increase up to 0.5 at 120 km. This value was constant up to 220 km, then there was slight decrease to 0.4 in PoD for the outer-most radiuses. FAR, the opposite behavior, started near 0.40 and maintained these values until 50 km, where a peak occurred, followed by gradual decrease to around 0.2–0.3 at 120 km, remaining constant up to 220 km, and ending with a slight increase. Additionally, there was larger difference between SPROG and extrapolation nowcast of around 0.05 for all radiuses up to 170 km. Figures 20 and 21 show the average PoD and FAR across distance rings per time after the start of the rain events, for each nowcast model. The averages were obtained for 30 km rings for these graphs for clarity. These graphs show that the PoD was low for rings closest to the radar and increased as the distance from the radar increases, and the opposite occurs for FAR. The slight decrease in the models' skill in the outer rings that was observed in Figure 19 can also be observed in these graphs, especially after 120 min after the start of the events. Additionally, the PoD tended to increase with time and the FAR decreased with time.

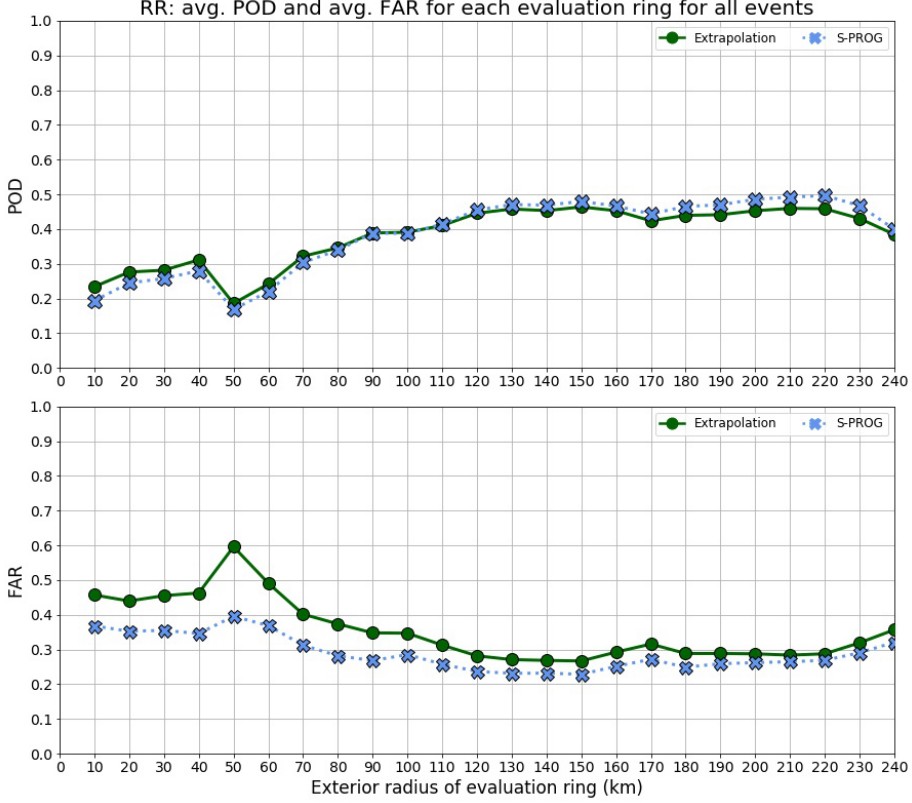

**Figure 19.** Average PoD (**top**) and average FAR (**bottom**) for all radar rain rate results per outer radius of the evaluation rings for each nowcast model.

The last two figures showed that between 100 km and 220 km, the nowcast performed consistently; this area includes the 180 km radius where the radar has the best "view" (no obstruction or problems with visibility) used in Section 3.2.5. And the nowcasts were less accurate, both at closest and farthest away points from the radar.

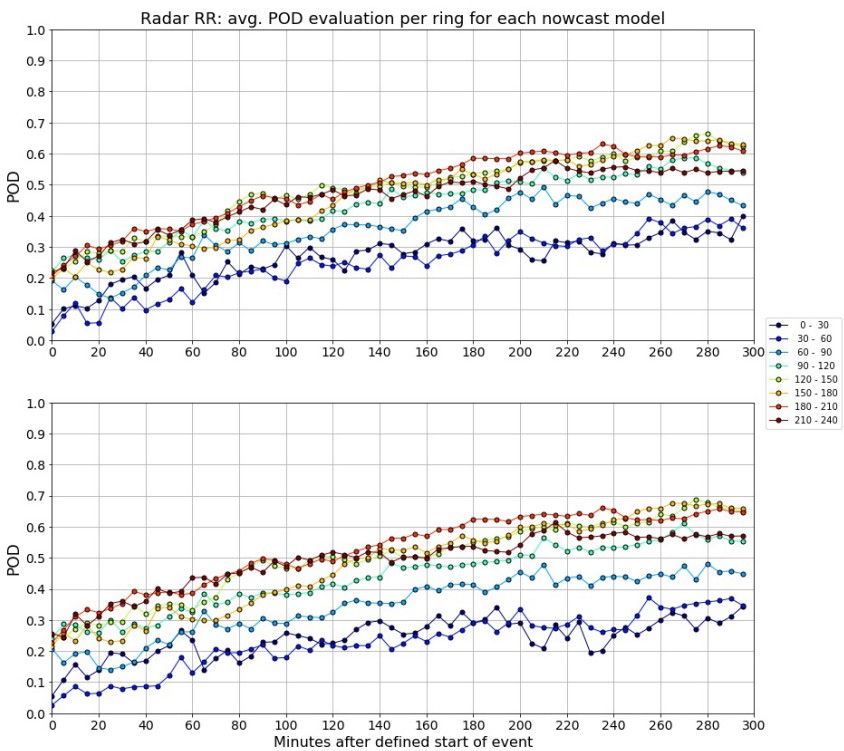

**Figure 20.** The change in average PoD over the period of analysis for 30 km evaluation rings for rain rate. The top graph is for extrapolation model and the bottom graph is for the S-PROG model.

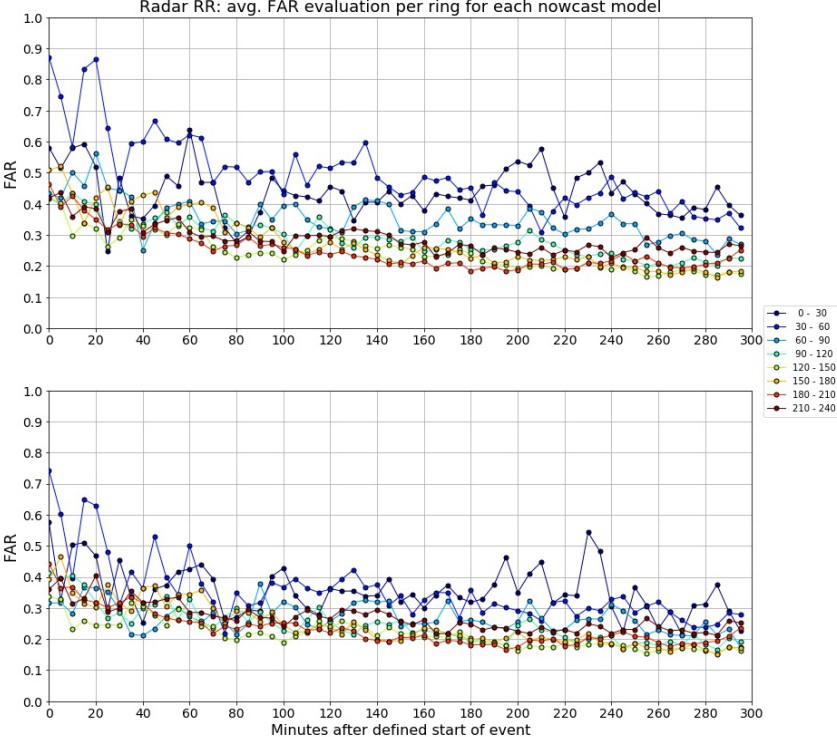

**Figure 21.** Same as Figure 20 for FAR.

### 3.2.8. Evaluation of the Intensity of Rain Rate Forecast by the Nowcast Models

The previous sections presented the nowcast evaluations using dichotomous verification statistics, which were used because the intensities of the satellite products are not expected to be correctly predicted by the nowcast models developed for radar data. Nevertheless, to complete the evaluation of the nowcast models for the rain rate, a brief

analysis on how both nowcast models predict the intensity of the rain rate provided by the Queretaro weather radar was performed. Figure 22 shows the average mean error (ME), mean absolute error (MAE) and the root mean standard error (RMSE) for each model, all obtained through the pySTEPS functions, for the 15-min forecast obtained every five minutes for the 295 min after the start of the events. The ME indicates that the models tended to underestimate more on average the rain rate for the first 165 min. The MAE showed that mean magnitude of the errors was generally below 4.5 mm/h for Extrapolation and decreased to around 3 mm/h at the end of the period of analysis, indicating that the errors in intensity tended to decrease as the events grew and evolved. A similar behavior was observed for S-PROG, but the errors were around 0.5 mm/h higher. Finally, the RMSE for Extrapolation started around 8 mm/h, and increased to approximately 11 mm/h at 160 min after the start of the events, followed by a decrease to around 9 mm/h by the end of the analysis. A similar pattern was observed for S-PROG with an increase of 0.5 mm/h. The results in comparison with the work of [8] are far better. The authors of [8] analyzed 10 events in Texas and obtained an MAE of between 11 mm/h and 12 mm/h on average. Additionally, the authors of [11] tested both S-PROG and Extrapolation among other models and found that the MAE was higher for S-PROG than Extrapolation, although [11] results were obtained comparing radar reflectivity and were presented as an average for various minimum reflectivity thresholds.

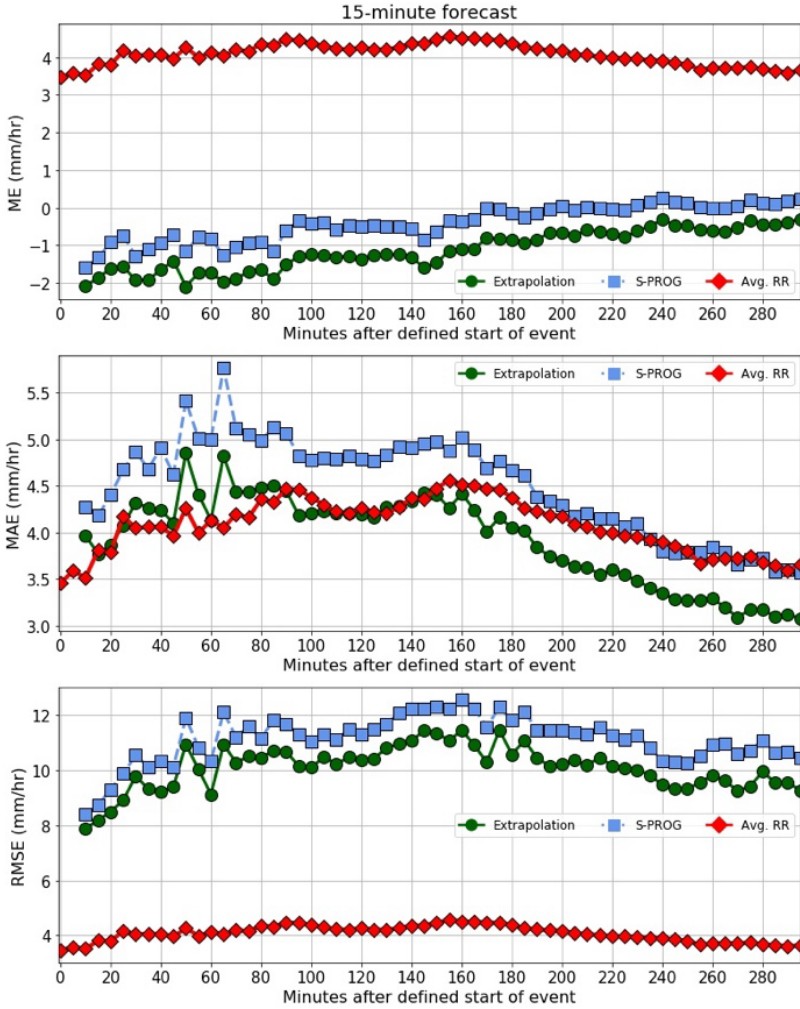

**Figure 22.** Average ME (**top**), MAE (**center**) and RMSE (**bottom**) for each 15-min forecast, obtained using the images in every 5-min interval as input image, for the Extrapolation and S-PROG nowcast in the 295 min after the start of the events. The red line indicates the average mean rain rate that occurs in that interval (average of the whole image observed in the interval).

Figure 23 shows the difference in the average of the mean rain rate for the Extrapolation, S-PROG and the observed rain rates. It is clear from these images that both models consistently underestimated the rain rate, between 2.0 mm/h and 1.2 mm/h for the Extrapolation model, and between 1.4 mm/h and 0.6 mm/h for the S-PROG model.

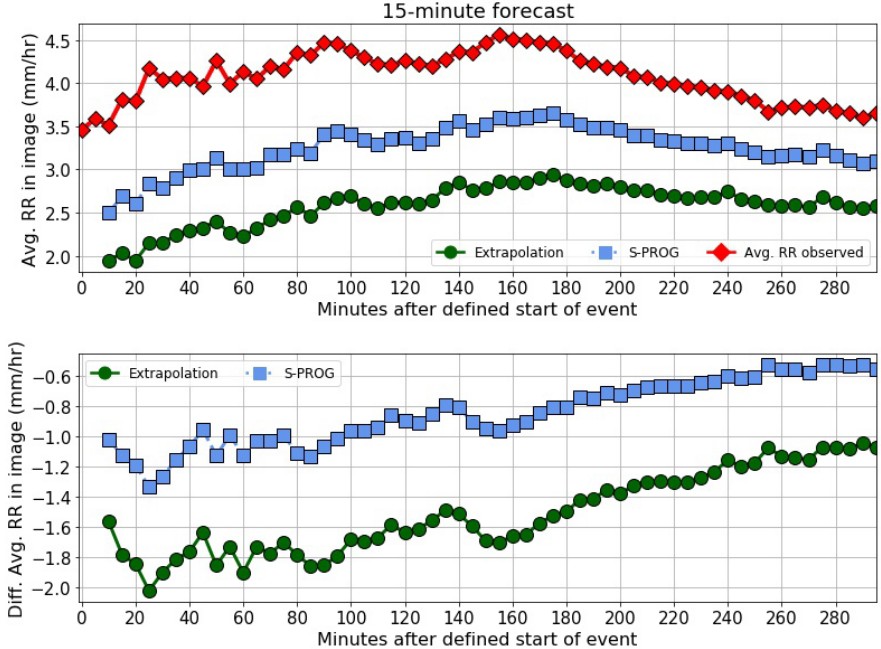

**Figure 23.** Average mean rain rate for the output image of the 15-min forecast, obtained using the images in every 5-min interval as input image, for the Extrapolation and S-PROG nowcast in the 295 min after the start of the events; and the average mean rain rate of the image observed in the interval.

## 4. Discussion

Due to the extensive number of results, a brief flowchart is presented in Figure 24 that highlights the main findings from the results. These findings are discussed in detail in this discussion.

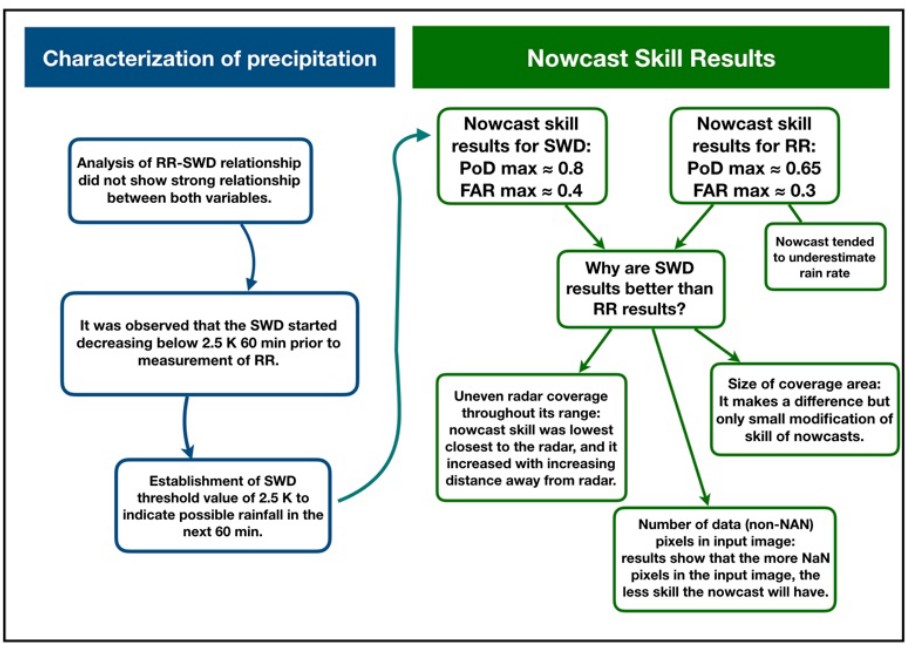

**Figure 24.** Flow chart of the main results obtained in this study.

The results of the evaluation of nowcast models suggest that the Extrapolation and S-PROG models can be used with the SWD data without any technical issues. In general, the 15-min forecast gave good accuracy, with the average accuracy being above 60% within the 90 min before and the two hours after the start of a precipitation event. Additionally, the utilization of the DATing module provided by pySTEPS can also be used to detect and track large clusters of the SWD data, albeit with some missed medium-size clusters. And due to its nature and how it works, the results of the 15-min forecast from the nowcast models when using the detected and tracked SWD clusters were far better than when using the entire SWD field available. As mentioned in the results, the only use of the pySTEPS library involving satellite data were primarily centered on the implementation of optical flow algorithms [46]. But the PoD and FAR results of the SWD data are consistent with previous studies [8,11,12] that used radar data for the evaluations of both models. However, a direct comparison cannot be made, given the distinct characteristics between SWD and radar data. As mentioned previously, the 15-min forecast was considered the shortest lead time that could have some operational used in the future, and the results obtained with this forecast were considered good. However, it is more ideal to use longer lead times with the SWD data set because the SWD dropped below a threshold value in the 60 min before the rainfall measurement. Considering this result, if the SWD is observed to decrease below a threshold value (2.5 K), one could potentially predict the possibility of rainfall up to one-hour prior. Thus, a 30-min or a 60-min forecast would be even more useful when using SWD data. For this reason, the evaluation analysis with SWD data was also performed using the 30-min forecast and the 60-min forecast. The general patterns observed with the 15-min forecast were also present with the 30- and 60-min forecast, including the differences in accuracy between using all the SWD field and only using the detected and tracked SWD clusters and the similar results from both nowcast models. However, the 30-min forecast was, on average, 15% less accurate than the 15-min forecast, and the 60-min forecast was around 30% less accurate, as expected because larger lead times result in less accurate forecast. Although these forecasts were less accurate, they can still be useful. Applying the methodologies used in this study, the nowcasts have the potential to predict the areas of possible rain development one hour prior to the event with an accuracy of around 40% and 30 min prior with an accuracy of approximately 50%. Nevertheless, further research is essential to enhance and refine these forecasts.

Furthermore, while the SWD fell below 2.5 K around 60 to 90 min before the rainfall event, not all areas of SWD that fell below 2.5 K developed into rain. These observations indicate that the SWD is not enough to detect a precise area of future rain development. This study only used the SWD because it lacked any other tools and other sources of data that could provide more information on the atmospheric condition of the whole area of study. As mentioned in the introduction, Mexico lacks a nationwide rain gauge system or a comprehensive meteorological station network, with the ones that exist being very scatter around the country to provide the necessary spatial resolution for these types of nowcast systems to function efficiently. Moreover, the authors also explored the use of other GOES-16 satellite products but found several to be unavailable at in the region in the period of analysis, and the ones that were available did not have a clear relationship with rain rate or the development of rainfall like the SWD did. Regardless, once the data sets for the other GOES-16 products become available for the region, future research can use these satellite products to further examine the precipitation that occurs in the region with the radar data, as well as determine more precise locations for future precipitation along with the SWD.

Continuing, the statistics used to evaluate the models were purposely chosen as dichotomous evaluators that only analyzed whether the forecast correctly predicted the presence of the SWD and not its magnitude. This is because these models were designed to predict radar derived rainfall rate or radar reflectivity, rather than satellite products. As highlighted in Section 3.1, the SWD tends to decrease before the presence of rain. Thus, if the models are predicting in general the dissipation of rainfall by decreasing the rain rate,

they can possibly cause an apparent increase in the development of convection when used with SWD values. For this reason, other nowcasts available in the pySTEPS library were not used for this work. The Extrapolation and S-PROG models yielded satisfactory results due to the absence of elements in the algorithms that directly alter the intensity of the input variable, like other models such as a dissipation factor. Additionally, the fact that S-PROG and extrapolation yielded similar results indicates that the spectral decomposition of the spatial scale used in S-PROG does not improve or modify the position forecast when using SWD, although a modification in the intensity was observed in the SWD forecast.

When using the radar rain rate data, the Extrapolation and S-PROG models demonstrated satisfactory performance, showing minimal disparity between using the entire rain rate field and solely considering the data within the SWD detected and tracked clusters. In addition, there was a significant increase in the skill of the nowcast as the events evolved and grew. The forecast obtained using the radar rain rate data was also less accurate than the forecast from SWD data, with the PoD being between 0.2 and 0.4 smaller for rain rate data. Conversely, the FAR was around 0.1 smaller for rain rate data. The slightly better results for FAR were mainly because the nowcast models tended to create more false alarms around the SWD data than the rain rate data, affecting statistics that depend on this variable, including FAR. Comparing the findings of this study with the earlier works of [8,11,12], the PoD and FAR values did not significantly deviate for developed events. Although the PoD values were significantly worse than those observed in both studies at the beginning of the events, within the events developed, the PoD improved, coming within approximately 0.1 of the values observed in the other studies. It is essential to note that the studies did not utilize the same threshold rain rate value (indicating rainfall and non-rain conditions.

Regardless, the overall results were less accurate than those obtained using SWD data. This discrepancy can be attributed, in part, to the higher presence of real (non-NaN) SWD data per data matrix compared to the rain rate data. As demonstrated in Section 3.2.6, a positive correlation exists between the quantity of real data pixels and most of the statistics. This correlation arises due to the increased impact of outliers on the results in instances of a reduced number of actual data points within an input image. Throughout the majority of each analyzed event, the SWD data substantially outnumbered the rain rate data, contributing to the superior statistical outcomes observed for the SWD. It was calculated that an image with at least 420 pixels (out of 58,880 total pixels) with rain rate values above the set noise threshold of 0.5 mm/h was necessary to obtain a skilled forecast. Another reason the radar rain rate results were not as good as the SWD results is the radar's lack of consistent accuracy throughout the radar coverage area, whether this is due to blockages from mountains or radio antennas or attenuation. For the Queretaro weather radar, most of the echo blockage occurs beyond 180 km radius. The effects of this were seen with the statistics with respect to the distance away from the radar were obtained, and the PoD and FAR showed the nowcast models' skill changed with the distance. In particular, the statistics were lower for distances close to the radar and at the edge of the radar coverage area. While the area between had consistent results on average.

Finally, in regard to evaluation of how the nowcast predict the intensity of the rain rate, the results were not very favorable. The MAE showed the average error was less than 5.5 mm/h, the RMSE was above 8 mm/h and the Extrapolation nowcast had smaller average values than S-PROG. However, the MAE and RMSE values were higher than the average observed mean rain rate (excluding 0.0 mm/h measurements), although they were similar to the MAE obtained by [11] and far better than those obtained by [8], whose MAE was twice as high. Upon comparing the average mean rain rage of the models and the observed values, the results showed that the models underestimated the rain rate. These results were not ideal, especially because it was assumed that the spectral decomposition of spatial scales and autoregression of the S-PROG would result in better RMSE scores, as observed in [1]. However, when considering all other results, the values obtained were more consistent with other previous studies [8,11,12], especially for developed events (approximate 120 min after the defined start of the event). Regardless, the fact that the

average mean rain rate was underestimated in a consistent manner for both models points to a possible path for resolving the issue. For this reason, it is crucial to continue studying these nowcast models for this region, to assess why these models are not performing well in forecasting the intensity of precipitation and how they could be improved.

## 5. Conclusions

This study explored the potential of the SWD derived from the GOES-16 satellite data as a possible indicator of rainfall. It was found that the SWD, on average, had a value of 1.99 K during rainfall. Furthermore, it was observed that the SWD typically decreased from around 3.8 to 1.99 K 240-min prior to the measurement of rainfall, and from 2.5 to 1.99 K 60 min prior to the rainfall. These results led to the establishment of the minimum threshold value of 2.5 K, which indicates a possible development of rainfall within the next 60 min.

Using the 2.5 K threshold, the SWD fields were used in two nowcast models provided by pySTEPS (extrapolation and S-PROG) to evaluate how this variable performed when used with nowcast models developed to be used with radar data, using the entire SWD field and only using the large clusters detected when using the pySTEPS DATing module, to attempt to obtain a forecast prior to the development of rain. The results showed that the S-PROG and extrapolation models, both of which did not significantly modify the intensity of the input data, performed well with the SWD data, and the performance improved for some cases as the start of the rainfall event got closer and thereafter. The 15-min forecast was 15% more accurate than the 30-min forecast and 30% more accurate than the 60-min forecast. Additionally, the DATing module algorithm was able to detect and track SWD clusters, but it did miss several clusters, which made the overall results of the evaluation with clusters less reliable than when using all SWD data available.

Finally, when the same models were evaluated using radar data, the results showed less accurate forecast than those obtained with SWD data, although the skill improved as the precipitation events developed. This was partly caused by the fact that the radar did not have the same accuracy over all its coverage area, and the SWD fields had more real data (non-NaN values) throughout the field. A decrease in the amount of real data available led to more pronounced effects of the errors and outliers on the utilized statistics. Furthermore, the models' ability to accurately predict the intensity of rain rate was not high, with the MAE being as high as the average rain rate in the region, the models underestimating average rainfall, and Extrapolation models having more skill in predicting intensity than the S-PROG model.

In conclusion, the extrapolation and S-PROG models exhibit skilled predictions up to 60-min, for the region of central Mexico. Furthermore, it was noted that the SWD can be utilized to identify potential areas of future rainfall. However, the SWD alone was not enough to have high precision in the prediction of rainfall, and the addition of other measurements such as wind or surface measurements is necessary the improve the detection areas of possible rainfall. Lastly, the methodologies used in this work can be used as a foundation to explore the use of other GOES-16 satellite products in the nowcast models and functions provided by the pySTEPS library. Further research to this study includes the improvements of the predictions by the nowcast models, for both radar rain rate and the satellite products.

**Supplementary Materials:** The following supporting information can be downloaded at: https://www.mdpi.com/article/10.3390/atmos15020152/s1, Figure S1. Rain rate from the Queretaro radar and SWD field below 2.5 K from the GOES-16 satellite for 12 events. Images present 1 h before start of the event, at the start of the event and 1 h after start of the events.

**Author Contributions:** Conceptualization, D.I.-F. and A.M.; Formal analysis, D.I.-F.; Investigation, D.I.-F.; Methodology, D.I.-F. and A.M.; Supervision, A.M.; Writing—original draft, D.I.-F.; Writing—review and editing, A.M. All authors have read and agreed to the published version of the manuscript.

**Funding:** This research received no external funding.

**Institutional Review Board Statement:** Not applicable.

**Informed Consent Statement:** Not applicable.

**Data Availability Statement:** Publicly available datasets were analyzed in this study. These data can be found here: https://www.avl.class.noaa.gov/saa/products/welcome (accessed on 10 April 2020. Data were also requested and obtained from the State Water Commission of Queretaro, Mexico: https://www.ceaqueretaro.gob.mx (accessed on 1 August 2020).

**Acknowledgments:** The authors would like to thank the Consejo Nacional de Ciencias y Tecnologia (CONACyT) for the support during the research of this paper. This work is part of the UNAM Project IA101219 "Estimación cuantitativa de la precipitación a partir de los datos del radar de Cerro Catedral".

**Conflicts of Interest:** The authors declare no conflict of interest.

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
