# Peer review of "Assessing Nowcast Models in the Central Mexico Region Using Radar and GOES-16 Satellite Data"

_atmosphere, doi:10.3390/atmos15020152_

Round 1

Reviewer 1 Report (New Reviewer)

Comments and Suggestions for Authors

Summary

This study explored the use of the SWD obtained using the GOES-16 satellite data as a possible indicator of rainfall using two nowcast models provided by pySTEPS (extrapolation and S-PROG).

Further investigations could be interesting to strengthen the methodology used, for example by using other satellite products and applying the nowcast models exposed in this work in different study areas.

Overall, I felt the paper was well-written and interesting, and I believe only minor revisions are needed. However, I have the following comments I think the authors should address before this paper is accepted for publication.

Comments

Line 47 often used threshold -> often used as threshold.

Line 89 “[FALTA]” reference is missing?

Line 103 Which version of pySTEPS?

Line 123 The evaluation of these methods used the dichotomous verification statistics -> The evaluation of these methods is based on the dichotomous verification statistics.

Line 144 Please rephrasing last period to clarify.

Line 161-165 Are there scientific references anywhere which use these thresholds for the SWD product? If affirmative, please add reference there.

In Figure 8 it should be useful to add a basemap.

Line 258-262 Are there scientific references anywhere which use these parameters? If affirmative, please add reference there.

In Figure 13 it should be useful to add a basemap.

In Figure 17 it should be useful to add a basemap.

Author Response

Thank you for your comments and critiques. They have been extremely helpful in improving the overall quality of our work. We have addressed and corrected all the errors you mentioned in your comments (all corrections from your review are highlighted in blue/cyan).

Comment 1:Line 161-165 Are there scientific references anywhere which use these thresholds for the SWD product? If affirmative, please add reference there.”

Response: The values were deduced from the results in section 3.1 and there are no references that use these threshold values with the GOES-16 data. However, we did add a reference that investigated the SWD using GOES-16 data for a case study in Northern Mexico near the US border. They observed similar changes in SWD (decreasing to values below 4 degrees Celsius) before and during the development of convection, but they did not derive any threshold values.

Comment 2: “Line 258-262 Are there scientific references anywhere which use these parameters? If affirmativeplease add reference there.”

Response: We were unable to find any references that used the SWD data from the GOES-16 satellite together with pySTEPS. These values were obtained through test and observation. 

All other comments have been addressed directly in the manuscript. This includes the addition of basemaps to the mentioned figures.

Reviewer 2 Report (New Reviewer)

Comments and Suggestions for Authors

Report on manuscript atmosphere-2642831

Assessing Nowcast models in the central Mexico region using radar and GOES-16 satellite data

In this manuscript, authors used the Python pySTEPS library - nowcast models. Authors used data of the radar rain ratio and the satellite product Split-Window Difference in central Mexico. Authors concluded that the python library can provide useful rain forecast. The paper has the potential to be published but after revision. I would like that the authors address the following issues:

Main Comments:

-        My main comments about the manuscript are:

o   Reading in figures 5,6 and 7, it seems that SWD is not very helpful to the rainfall intensity. Authors tried to add more explanation about this point in their paragraph L226, however, this doesn’t seem good enough to consider a valid relationship between SWD and rainfall intensity. Please explain why is it significant to consider SWD as a powerful factor!

o   Figure 11, 12, 15 and 16 are misleading, you need to zoom in to see the real behaviour of these graphs. You left 90% of the plot empty.

o   Statistical evaluation is missing. This is important to confirm the validity of these results. If not, we need to motivate how can we trust them.

o   Authors need to reread their manuscript and improve the English language, I highlighted few issues below, but more extensive revision is needed to improve the readability of the paper.

More comment:

-        Please motivate why central Mexico is left behind and no nowcasting models are used to study this location.

-        Nowcasting models are a good tool… à are good tools …

-        Add a reference for the nowcasting models in the first paragraph

-        L.33: one is located ….., and the third is located

-        L40: pySTEPS library à add a reference

-        What is the reason behind choosing this data sample from NOAA’s GOES 16 satellites between July 1st and August 16th in 2018.

-        L.86: Lat à Lat… Lon à Lon.

-        List of abbreviation can be added as Appendix at the end of the manuscript.

-        L95 and L96: rephrase, also you need to motivate the seclection of the 2x2 km grid.

-        L163: How and where do you read this difference 37.2% and 82.2%?

-        L202: threshold is at 2.5K, why?

Comments on the Quality of English Language

I highlighted few issues in my report, and authors need to conduct another check to improve the English of their paper

Author Response

Thank you for your review and comments, they were instrumental in improving the overall quality of our work. All English errors have been corrected (corrections to your comments are highlighted in green).

For the individual comments:

Comment 1: Reading in figures 5,6 and 7, it seems that SWD is not very helpful to the rainfall intensity. Authors tried to add more explanation about this point in their paragraph L226, however, this doesn’t seem good enough to consider a valid relationship between SWD and rainfall intensity. Please explain why is it significant to consider SWD as a powerful factor!”

Response: We agree with your observation that there is very little correlation between the SWD and the rain rate. However, the objective to SWD is to predict the FUTURE development of rain fall. We have added one more reference where the SWD from the GOES-16 was shown to be a good indicator of future convection (reference 30). The ultimate objective for choosing the SWD is to obtain nowcast predictions PRIOR to rainfall, subsequently, the rain rate from the radar can be used in the models. The reason we included the SWD forecast during precipitation events was to enhance our results and have better statistical analysis. We have further clarified this point in section 3.2.1 (L224).

Comment 2: Figure 11, 12, 15 and 16 are misleading, you need to zoom in to see the real behaviour of these graphs. You left 90% of the plot empty.”

Response: The y-axis had values from 0 to 1 because that's the range for most of the statistics we used. We believe that for POD, FAR and HSS, it is important to show where the values fall within the whole range (0 to 1) for a better understanding of the results; we want to know were these values land between 0 and 1, which might be harder to see if the graphs are zoomed in to a specific interval. It is also helpful when comparing the statics with each other. For instance, in our study, we often compare the results of FAR and POD; with figure 11, we can easily compare both variables, observing the fact that the average POD is higher than the average FAR, given that both graphs have the same Y-axis range. While we understand your concerns, we believe that keeping the y-axis with the range 0 to 1 is best for the overall work.

Comment 3: “Statistical evaluation is missing. This is important to confirm the validity of these results. If not, we need to motivate how can we trust them.”

Response: We're unclear about your comment “Statistical evaluation missing.” This point seems broad, making it challenging to pinpoint the specific concern. The majority of our presented results refer to the statistical analysis done with the nowcast results obtained. Do you believe we need more than the dichotomous results? We opted for these specifically, because the models we used cannot correctly modify or predict the intensity of the SWD, given that they are not built for this particular variable, Our aim was to determine if they can predict the location of the variable (in other words, if the model could accurately forecast the presence of SWD in a given pixel of the grid). Six other researchers have independently reviewed this work and none have mentioned any problems with the statistical analysis, so it would be helpful to expand on what the issue is in order to fix it and improve our study.

Comment 4: Authors need to reread their manuscript and improve the English language, I highlighted few issues below, but more extensive revision is needed to improve the readability of the paper.”

Response: We have thoroughly proofread he work and implemented the necessary corrections. We believe the content is more accessible and readable.

Comment 5: Please motivate why central Mexico is left behind and no nowcasting models are used to study this location”

Response: No documents or studies specifically address the absence of nowcast models in Mexico. We believe the limited research in this area derives from the lack of funding, tools/infrastructure and interest. We do not believe it is appropriate to expand on these issues without solid evidence or justification within the scope of this study.

Comment 6: L95 and L96: rephrase, also you need to motivate the seclection of the 2x2 km grid”

Response: We chose the 2 x 2 km grid because its rectangular shape is easier to use in pySTEPS (this was clarified in the manuscript).

Comment 7: “What is the reason behind choosing this data sample from NOAA’s GOES 16 satellites between July 1st and August 16th in 2018.”

Response: We selected those dates based on the availability of radar data during the rainy season. (this was clarified in the manuscript).

Comment 8: “L202: threshold is at 2.5K, why?”

Response: The 2.5 K was determined from graph 8 (this was clarified in the manuscript). In this graph, the average SWD tended to decrease below 2.5 K one hour before the development of rainfall.

All other comments were addressed in the manuscript (green highlighter)

Reviewer 3 Report (New Reviewer)

Comments and Suggestions for Authors

This paper goals were to evaluate the SWD as an indicator of rainfall and to use SWD and radar data as input to test two nowcast models provided by pySTEPS library.

The use of SWD is interesting as it could indicate the convective activity before rainfall. In general, the results were reasonable but lacked a discussion based on the literature which makes it difficult the understand of its results' significance. 

- In the introduction, one or two paragraphs with a review of papers that evaluate these models around the world would help the reader to understand the importance of this work. Results obtained by those could also help with the discussion of the results.

- Check line 89 [FALTA]. 

- In the methodology, which days were removed from the data set?

- In Figure 1,  a zoom from a broader area would make it easier for the reader to understand where the area is located.

- Is it necessary to keep figures 1 and 2?

- Did you only use radar and satellite data to establish the rain time of occurrence and intensity?

- In the results, a simple description of the rain registered throughout the study period could help.

- A discussion of the results based on the literature is necessary. There are almost no references throughout the results and discussion section.

- Figure 5 resolution needs improvement.

- The difference between RR_aa and SWD_aa are significant. What are the possible reasons for this behavior?

The results of Figure 8 are interesting but I would like to see more examples. It is difficult to conclude something from a single example. Maybe others could be added as supplementary data.

- How did you calculate the POD, FAR, and FA? What are the values of those indices for intense and extreme rainfall? This would be an interesting analysis.

- It is very difficult to analyze figures 22 and 23. Maybe reducing the number of rings could help with the visualization. 

- As a suggestion, since there are a lot of results in this paper, a flowchart could be used at the beginning of the discussion section as a way to summarize and highlight the main results.

- Are all figures and sections really necessary for the full understanding of the paper? Maybe some of it could be shown as supplementary material. 

Author Response

Thank you for your review and feedback, they were very helpful in further improving the quality of our work. All modifications based on your comments are highlighted in yellow.

Comment 1 and 2: The use of SWD is interesting as it could indicate the convective activity before rainfall. In general, the results were reasonable but lacked a discussion based on the literature which makes it difficult the understand of its results' significance.” and “In the introduction, one or two paragraphs with a review of papers that evaluate these models around the world would help the reader to understand the importance of this work. Results obtained by those could also help with the discussion of the results.”

Response: We have incorporated additional references regarding the use of pySTEPS models with radar data, as suggested, we've also added references to results in the discussion section. We could not find any references linked to pySTEPS and the satellite data or pixel analysis. For the SWD results, we mentioned some references that are solely related to radar data and not SWD. Your feedback helped correct a big oversight in our analysis, so thank you.

Comment 3: “In the methodology, which days were removed from the data set?”

Response: The specific days were added in the methodology. The days are July 2nd, 17:42Z to July 5th, 17:42Z, July 16, 2:57Z to July 17, 14:07Z, and August 16 starting at 2:02Z.

Comment 4: “Is it necessary to keep figures 1 and 2?”

Response: We believe emphasizing where and what is being used to obtain the results is crucial.

Comment 6: “Did you only use radar and satellite data to establish the rain time of occurrence and intensity”

Response: Yes, as mentioned in the manuscript, we only have access to these two data sets for the entire study region. Other rain data sets such as rain gauge data were limited because they cover a very small area (near the center of the grid) corresponding to the city of Queretaro and and provide only hourly or daily values. We utilized the best available data for our analysis.

Comment 7: “In the results, a simple description of the rain registered throughout the study period could help.”

Response: The type of precipitation that occurs is convective precipitation. We've included this detail at the beginning of the methodology section.

Comment 8: The difference between RR_aa and SWD_aa are significant. What are the possible reasons for this behavior?”

Response: The behavior observed in the preceding section indicated that the SWD tended to decrease with the presence of rain rate (even though the relationship between SWD and rain rate intensity is relatively weak). Additionally, the starting increase in the SWD at 6 hrs local time aligns with sunrise, indicating that sunlight reduces low-level water vapor. We have expanded on this observation, providing more clarity and details, in section 3.1.1."

Comment 9: “The results of Figure 8 are interesting, but I would like to see more examples. It is difficult to conclude something from a single example. Maybe others could be added as supplementary data.”

We have images for all events, and we will try to include more as supplementary data.

Comment 10: “How did you calculate the POD, FAR, and FA? What are the values of those indices for intense and extreme rainfall? This would be an interesting analysis.”

Response: We have added the equations used to calculate POD, FAR and FA (they were calculated using pySTEPS, which uses the most common definition of each).

Your suggestion of analyzing extreme events is something we are very interested in. However, we feel like it is extremely important to do a full and thorough analysis for extreme events for this region since it is a high frequency occurence. For this reason, it is not ideal to add it to this particular analysis due to the existing depth and breadth of the content, we believe the results could be lost if added here. Our primary focus in this work is the application of nowcast in a region where it has not been previously studied.

Comment 11: “As a suggestion, since there are a lot of results in this paper, a flowchart could be used at the beginning of the discussion section as a way to summarize and highlight the main results.”

Response: We find this to be a good idea and have incorporated the flowchart.

Comment 12: “Are all figures and sections really necessary for the full understanding of the paper? Maybe some of it could be shown as supplementary material.”

Response: We do believe all the elements are essential. However we did remove the graphs of the statistics we don’t mention so frequently. (FA, ACC and BIAS).

Reviewer 4 Report (New Reviewer)

Comments and Suggestions for Authors

A synthetic approach regarding the use of satellite data and rainfall forecast models. The file is attached. 

Author Response

Thank you very much for your comments and review. We have made some changes and added some references (highlighted parts) and we feel the work is much better.

Just to clarify, the highlighted sections on the manuscripts you read were placed there because it was a resubmitted work (this is the second submission of this work to the journal). We were asked by the journal to highlight all the changes we made to the very first manuscript we submitted in case the original reviewers wanted to read the manuscript again. The new revised manuscript we are submitting now has other sections highlighted to point out the parts we modified or added to the manuscript you read in this round of peer review. 

Round 2

Reviewer 2 Report (New Reviewer)

Comments and Suggestions for Authors

The authors have satisfactorily addressed my comments. I recommend accepting the manuscript in its current form.

Reviewer 3 Report (New Reviewer)

Comments and Suggestions for Authors

I do not have any extra comments. 

This manuscript is a resubmission of an earlier submission. The following is a list of the peer review reports and author responses from that submission.

Round 1

Reviewer 1 Report

Comments and Suggestions for Authors

This study mainly evaluated the nowast models pySTEPS using radar rain rate and SWD data. I have some questions about the paper. 

1) What is the reason for chooing SWD? According to Figure4, it is hard to seprate precipitaiton from non-precipitation pixels;

2)Rain gauges usually have high quality rain rate observations, it would be better to validate the nowcasting results with rain gauge observations.

Comments on the Quality of English Language

None

Author Response

Dear reviewer,

Thank you for your comments. They were of help for us in improving the quality and understanding of our work. As for the questions you have regarding the paper,

  1. The reason for using SWD was that although it’s relationship with the intensity of precipitation was not clear, it’s change before the development of rainfall was clear. It was observed (figure 5) that the variable decreased from 2.5K to 2K on average in the 60 minutes prior to the rainfall. Thus, we used it mainly to extend the application of the nowcast to times prior to the rainfall. If the SWD fell below 2.5K, then the area could develop into precipitation. The radar data was then used along with SWD data when rainfall started. We corrected this in the paper by adding a paragraph on section 3.2 explaining this decision.
  2. The use of secondary precipitation data sets along with the radar is very useful to support or improve the results observed. However, Mexico does not have an extensive rain gauge network. There are small rain gauge networks in the big metropolitan areas that fall within the region of study (Mexico City and City of Queretaro) but these give hourly or daily measurements, which is not helpful for this type of study, and only cover a small percentage of the total area. In this study, we used all the meteorological data available to us for the entire region of study, which was radar and GOES-16 data. A mentioned of the lack of rain gauge system was added to the introduction.

Reviewer 2 Report

Comments and Suggestions for Authors

Review of “Assessing Nowcast models in the central Mexico region using radar and GOES-16 satellite data” by Diana Islas-Flores  and Adolfo Magaldi  etc.

This paper tries to access the nowcast models with radar data and satellite-derived Split-Window Difference (SWD) data. Unfortunately, this manuscript does not present the work well. My suggest is to reject its publication. 

First, the authors do not describe the method properly. Frankly, I cannot understand many terms in the paper and why they are presented, for example, “percentage of cases”, “SWD clusters”, “All SWD field” and many more. Furthermore, I do not get what are the input variables for the nowcasting models and what are the predictable variables? And how the evaluation scores of POD, FAR etc are computed because there is not verification (or observation) datasets are described.

Second, I do not get the values or meanings of the presentation of most of the figures in the “paper” and their descriptions. What are roles and meanings? How are they connected? It seems that they just appear suddenly without context. They does not support the conclusion and the discussion in the “paper”.

Finally, the organization of the paper is problematic. There are two section 5, one is for “discussion” and the other is for “conclusion”. Captions for some figure is meaningless (for example, figure 2) and some section have paragraphs without sense (for example Section 3).

The “paper” does not have any scientific soundness, and it also does not provides any practice steps for operational implementation, for examples, the step to retrieve satellite or radar data and preprocess step, the step to run the nowcast models and the step to retrieve the prediction and its verification etc. 

Comments on the Quality of English Language

The English seems good. I know every word but it is hard to understand the meanings of a lot sentences and how they are connected within the manuscript's context.

Author Response

Dear reviewer 2:

Firstly, we fixed all the clear typos you pointed out in the fourth paragraph of your “comments”.

Second, before continuing, we would like to make it clear that we take every comment, suggestion and criticism seriously and we have made all appropriate corrections based on suggestions we have received for this paper. However, we will not be doing any modifications to the paper based on your “comments” besides the typos indicated and a second proofread to correct any mistakes missed. We had this paper reviewed by five other people (4 were independent reviewers) and they did not have any issues with understanding all the terms, definitions and descriptions used in the paper. They understood the images and graphs and how they were connected to the text, including the discussion and conclusions of the work, and found that the research had scientific soundness. The English of the paper was also reviewed and found to be acceptable and comprehensible.  

Finally, the paper is not a step-by-step guide for operational implementation. It is clearly stated in the paper that it is an evaluation of the nowcasts in a region where nowcast are not being used in order to see if future operational implementation is possible. Its objective was to see if the nowcast can be used with the data available in the region and to assess how accurate they are with the data. Its objective was not to start an operational implementation. Nevertheless, this point was further clarify in the paper to remove any possible doubts.

Reviewer 3 Report

Comments and Suggestions for Authors

The manuscript is well written. could you compare and analyze 1-2 typical extreme precipitation?

Comments on the Quality of English Language

The language is fluent and accurate, without obvious language errors.

Author Response

Dear reviewer 3:

Thank you for your comments and review. Regarding your suggestion to compare and analyze extreme events, this is a good idea. We do analyze the average SWD with respect to rain rate intensity, with the extreme events. And we observed that the results of the rain rate nowcast forecast improved with the development of a convective system because it has more data pixels to work with (We added some comments in the paper to highlight this). However, we believe that this work is already extensive, and that adding a deeper analysis of extreme events will make the paper too large that some of the work might get lost. We will be extending the work of analysis of precipitation in the region using radar and more satellite products. Hopefully, more satellite products will be available for the region and we will used them to improve the analysis of the precipitation events and extreme precipitation events.  

Round 2

Reviewer 1 Report

Comments and Suggestions for Authors

I think the SWD is not a good indicator for precipitaiton as was shown in figure, especially for stratiform precipitation. The auther did not provding reasonable proof for the reason of choosing SWD.

Besides, the high POD (70%) and low FAR(40%) were not convincing if the rain gauge observations were not used in the evaluation. 

Reviewer 2 Report

Comments and Suggestions for Authors

The modifications are just several minor sentence changes. I still see no scientific value of this paper with respect to either research or operations.  It may also be limited by my knowledge with the nowcast model studied in this paper. The nowcast models studied in the paper is not a numerical model of my study for Numerical Weather Prediction. My review suggestion is rejection for publication.

Reviewer 3 Report

Comments and Suggestions for Authors

I think the manuscript has been sufficiently improved to warrant publication in Atmosphere.